# Tracking the global reduction of marine traffic during the COVID-19 pandemic

David March [1,4 ✉], Kristian Metcalfe [1], Joaquin Tintoré [2,3] & Brendan J. Godley[1]

The COVID-19 pandemic has resulted in unparalleled global impacts on human mobility. In the ocean, ship-based activities are thought to have been impacted due to severe restrictions on human movements and changes in consumption. Here, we quantify and map global change in marine traffic during the first half of 2020. There were decreases in 70.2% of Exclusive Economic Zones but changes varied spatially and temporally in alignment with confinement measures. Global declines peaked in April, with a reduction in traffic occupancy of 1.4% and decreases found across 54.8% of the sampling units. Passenger vessels presented more marked and longer lasting decreases. A regional assessment in the Western Mediterranean Sea gave further insights regarding the pace of recovery and long-term changes. Our approach provides guidance for large-scale monitoring of the progress and potential effects of COVID-19 on vessel traffic that may subsequently influence the blue economy and ocean health.

[1] Centre for Ecology and Conservation, College of Life and Environmental Sciences, University of Exeter, Penryn, UK. [2] ICTS SOCIB – Balearic Islands Coastal Observing and Forecasting System, Palma de Mallorca, Spain. [3] IMEDEA (CSIC-UIB), Mediterranean Institute of Advanced Studies, Esporles, Spain. [4] Present address: Centre for Ecology and Conservation, College of Life and Environmental Sciences, University of Exeter, Penryn, UK. ✉email: D.March@exeter.ac.uk

The coronavirus disease (COVID-19) pandemic has emerged as both a global health and socioeconomic crisis, with many countries implementing unparalleled mobility restrictions to control the spread of the virus. This unprecedented event, which has been referred to as the "anthropause", a period of reduced human mobility[1], has led to sudden and often dramatic reductions in transport, energy consumption and consumer demand resulting in significant changes in the scale and extent of human stressors and their associated impacts on the natural environment[2–9]. To better understand the potential effects on the blue economy and the environment, there is an urgent need to quantify the magnitude and patterns of the changes in human activities at sea.

Human activities in the ocean have been radically altered by the COVID-19 pandemic, with reports of port restrictions and changes in consumption patterns impacting multiple maritime sectors, most notably fisheries, passenger ferries and cruise ships[10–13]; sectors which rely heavily on the movement of people and goods. As with previous economic recessions[14,15], changes in vessel movement associated with COVID-19 are also likely to result in significant short- and long-term effects on multiple anthropogenic pressures, such as air pollution[15–18], the spread of invasive alien species[19,20], or collisions with marine animals[21,22]. Localised studies have already reported short-term declines in underwater noise[23], water turbidity[24] and fishing effort[11] as a result of the reduction of the vessel activity during the first wave of the COVID-19 outbreak. However, as mobility restrictions vary among countries and maritime sectors, the spatio-temporal effects of COVID-19 on ship-based activities and their influence on the marine environment are still unclear at global and regional scales.

Fortunately, recent technological advances associated with the Automatic Identification System (AIS), in particular the emerging constellations of microsatellites (S-AIS), now means that ship-based mobility patterns can be monitored globally at high spatio-temporal resolution[25–30]. AIS is a vessel identification system that transmits real-time information on routes of vessels via VHF radio transponders. AIS is required on all ships of 300 gross tonnage or more engaged on international voyages, all cargo ships of 500 gross tonnage or more, and all passenger ships, irrespective of size. In addition, individual countries may require AIS usage on additional vessels. For example, AIS is required for EU fishing vessels >15 m in length[31]. Moreover, AIS is also increasingly used on a voluntary basis by many other vessels, including smaller fishing and leisure craft, thereby providing a unique opportunity to monitor the location of a diversity of vessels across the world[28–30]. Despite some limitations of the system (e.g. small vessels not equipped with AIS, and transmission gaps[31]), historical and real-time AIS data have been shown to provide unparalleled insights into shipping-derived impacts and conservation planning at multiple spatial and temporal scales[32–35]. With regard to COVID-19, AIS has recently been employed to assess the potential spread of the virus[13,36,37] and to describe the reductions in marine traffic at local scales[11].

Here, we use AIS data to conduct a comprehensive assessment of the short-term changes on ship-based mobility patterns in response to COVID-19 across multiple sectors and at different spatio-temporal scales. First, we illustrate our approach by conducting a global assessment using monthly traffic density maps to evaluate changes in vessel activity across multiple regions and maritime sectors during the first half of 2020, relative to the same period in 2019. Then, we assess similar factors with high temporal resolution (i.e. daily basis) in the Western Mediterranean Sea, a key region for the global shipping network[38] and cruise tourism[39], which includes three European countries heavily impacted by the COVID-19 outbreak (i.e. Italy, Spain and France). Our approach quantifies the magnitude and patterns of changes in ship-based activities, providing data that can inform large-scale investigation of the potential socioeconomic and environmental effects of COVID-19 on the world's ocean.

## Results

**Global changes in marine traffic density**. To compare government measures among countries we used the Stringency Index from the Oxford COVID-19 Government Response Tracker (OxCGRT)[40]. Lockdown measures across coastal countries analysed ($n = 124$) started after the World Health Organization pandemic declaration on 11 March 2020, although China, the reported source of the outbreak, started to establish confinement measures by late January (Fig. 1). Overall, global confinement measures reached their maximum (i.e. strictest) levels during the month of April (Stringency index = $79.4 \pm 14.7$, mean ± SD; Fig. 1a and b), by which time China had started to ease lockdown restrictions (Fig. 1c).

We analysed global patterns in marine traffic during the first half of 2020 (January–June) using monthly density maps (at 0.25-degree resolution) from satellite AIS. An important characteristic of the AIS data is their stratification according to ship categories, thus allowing attribution of the spatial footprint of marine traffic to different maritime sectors. Merchant vessels (i.e. cargo and tankers) were widespread along major shipping lines, fishing and "other vessels" (e.g. service and recreational vessels) were more dispersed between coastal and offshore waters, while passenger vessels presented a more limited distribution (Fig. 2, Supplementary Fig. 1).

In order to assess potential disturbances of marine traffic in response to COVID-19, we quantified the absolute and relative changes of monthly density maps in comparison with the same reference period in 2019, thus accounting for general seasonal variability of ship-based activities. Average change in traffic density was unevenly distributed across the globe and varied by vessel category (Fig. 3, Supplementary Fig. 2). Changes in merchant vessels were differentially distributed across the major shipping lanes. Passenger vessels were most negatively affected in traffic density, especially in tourist hotspots like the Caribbean and the Mediterranean Sea. Conversely, changes in fishing and "other" vessels were more diffusely spread across the world's ocean (Fig. 3). Major changes in traffic density across all sectors were mainly found in coastal areas and the northern hemisphere, although fishing and "other vessels" also presented increases at greater distance from the coast (Supplementary Fig. 3).

Temporal variation of global changes in marine traffic density was assessed by estimating the occupancy and the proportion of grid cells with decreases and increases on a given month (Fig. 4). Considering all vessels together, we found that January and February presented general increases in comparison to equivalent months from 2019 (Fig. 4a). Then, as the pandemic was declared in March 2020, the proportion reversed, with 52.2% of cells presenting decreases in traffic density. This overall decrease remained at similar levels for the remaining 3 months of the study period, with decreases peaking in April (54.8% of cells). The patterns, however, were variable by sector. Cargo vessels presented decreases in January and this was maintained throughout the remaining period. Patterns for passenger and "other vessels" were similar, with more marked decreases after March, albeit the magnitude for passenger vessels being more severe. Reduction in tankers was not apparent until May and June. Fishing vessels, on the other hand, showed indications of recovery from May (Fig. 4a). Similar patterns were found in monthly changes of occupancy, calculated to reflect any shifts in the spatial extent used by marine traffic (Fig. 4b). Considering all

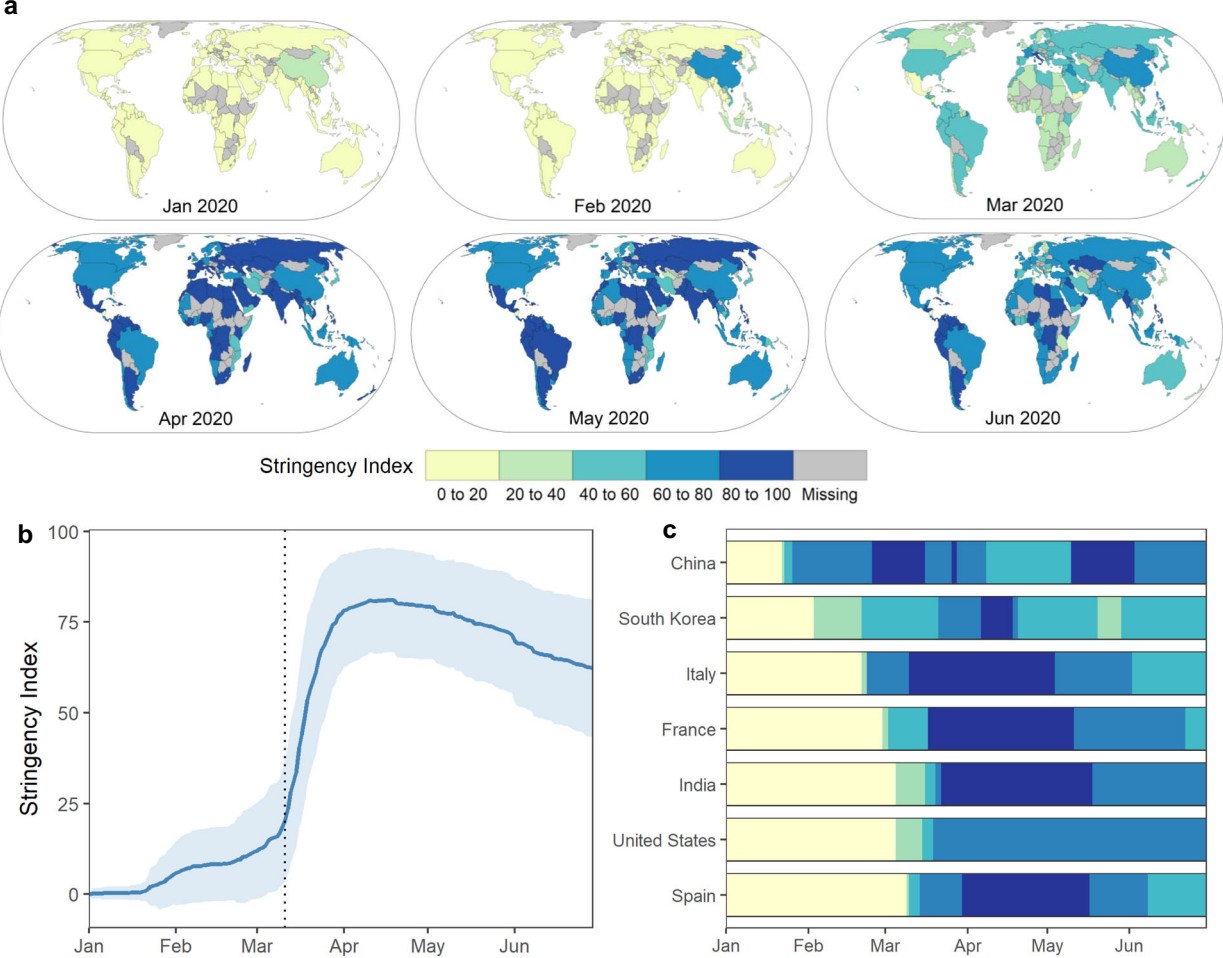

**Fig. 1 Spatial and temporal variation of the confinement measures in coastal countries during the first half of 2020.** We use the Stringency Index (100 = strictest response) as an indicator of confinement measures for coastal countries ($n = 124$). **a** Monthly mean per country. **b** Global daily average and standard deviation from 1 January 2020 until 30 June 2020. Vertical dotted line represents the World Health Organization pandemic declaration on 11 March 2020. **c** Individual series for selected countries, ordered according to the first date when the Stringency Index was above the first quintile. Note that data were not available for all coastal countries.

vessels together, there was a maximal reduction of 1.4% in their occupancy during April, with the largest drop in occupancy (30%) for an individual sector associated with passenger vessels.

Spatial and temporal changes varied regionally (Fig. 5). In European Seas, there was an almost universal decrease in vessel traffic after March 2020 (Fig. 5a), while patterns in other regions (e.g. East China Sea, Fig. 5b), and around some major shipping lanes (e.g. Arabian Sea, Fig. 5c) showed a mixed picture. Conversely, other regions showed overall increases in traffic density (e.g. Indonesia; Fig. 5d). At the local level, our analysis also captured profound decreases in traffic around some focal areas such as marine protected areas (e.g. Galapagos Islands in Ecuador, Fig. 5e) or in the vicinity of major ports (e.g. Port of Vancouver in Canada; Fig. 5f).

We further extended our analysis around 10 selected maritime chokepoints[41,42]. Maritime chokepoints constitute narrow passages that concentrate marine traffic, and thus are ideal locations to monitor synthetic changes in traffic density that may be otherwise masked. Temporal patterns were variable across chokepoints and between sectors (Fig. 6, Supplementary Fig. 4). After March, there were marked decreases in the Panama Canal, Strait of Gibraltar, Strait of Dover and the Bosphorus Strait driven by substantial changes in cargo and "other" vessel activity. In contrast, there were progressive increases in vessel activity in

some areas such as around the Cape of Good Hope driven by the growth of cargo traffic.

During the study period, there was an overall decrease in average traffic density in 70.2% of the Exclusive Economic Zones (EEZ) of the analysed countries ($n = 124$). Temporal patterns were heterogeneous among countries (Fig. 7). China presented major decreases during January and February and then exhibited an early rebound during March and April. Most countries, however, exhibited steep decreases after March. By June, several countries showed signs of recovery (e.g. USA, South Korea, India), while others continued at low levels (e.g. United Kingdom, Peru). We assessed the effects of confinement measures on marine traffic density at country level using linear mixed models (LMMs). We found a significant effect of the Stringency Index on the change of marine traffic density for all vessel categories, except for fishing vessels (Table S1, Supplementary Fig. 5). In addition, we found that the effect of lockdown restrictions was uneven across economies, with lower-income countries being less affected by confinement measures.

**Temporal changes in the Mediterranean Sea.** The Western Mediterranean Sea was one of the areas with the highest reduction in shipping activities (Fig. 5a). In order to capture short-term responses due to lockdown effects at finer temporal resolution, we

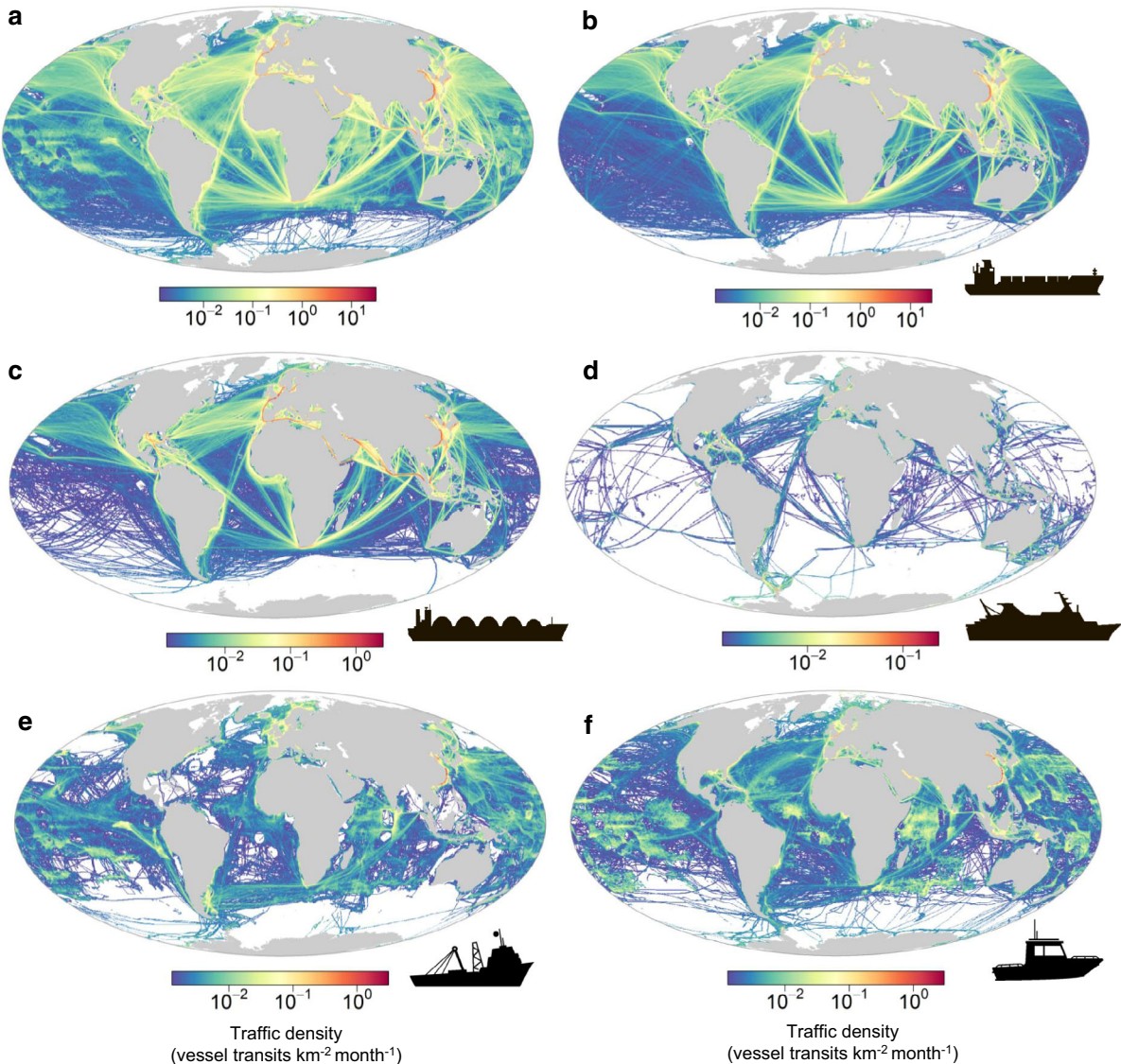

**Fig. 2 Global density of marine traffic.** Density maps showing the average number of vessel transits per square kilometre per month during the first half of 2020 (January–June). Vessel categories: **a** all vessels, **b** cargo, **c** tanker, **d** passenger, **e** fishing and **f** other vessels. Density estimates plotted on a logarithmic (log 10) colour scale.

used a real-time, long-term AIS dataset from land-based stations. In contrast to our global analysis, we considered an additional ship category, recreational boats, which constitutes an important sector in one of the world's tourism hotspots. To analyse the temporal variability of marine traffic during 2020 (1 January–30 November), we counted the number of vessels underway on a daily basis. The multi-annual distribution of the number of vessels in the Western Mediterranean was consistent through time for merchant and fishing vessels (Supplementary Fig. 6). Temporal variation showed a marked seasonality in passenger, recreational and "other" vessels, with a peak during the boreal summer, and a growing trend in the number of vessels across years. In 2020, daily counts of the number of vessels showed a significant reduction after the World Health Organization (WHO) declared a pandemic on 11 March, a pattern that was consistent across all sectors (Fig. 8, Supplementary Fig. 7). When compared to pre-disturbance baselines (i.e. equivalent periods of 2019), the number of vessels sharply decreased in the first days of mobility restrictions, reaching an overall median drop of 51% during the initial national

lockdowns, which lasted until approximately until 22 June, when countries from the study area (i.e. Spain, France, Italy) relaxed their confinement measures. Reductions were rapid and profound for all categories other than merchant vessels, for which reductions were not apparent until May. Maximal reductions ranged from 22.2% (tankers) to 93.7% (recreational boats), with a maximal overall drop across all categories of 62.2% during mid-April (Fig. 8, Supplementary Fig. 7). Passenger vessels presented the highest median drop (47.5%) during the first lockdown. Recovery rate was uneven among sectors. Cargo, tanker and, in particular, fishing vessels showed a relatively swift recovery in vessel activity. In contrast, passenger and recreational vessels remained at low levels throughout the lockdown period. After easing of lockdown restrictions, merchant and fishing vessels were close to pre-lockdown values. Recreational boats exhibited a fast recovery and rebounded their activity from mid-July until mid-September. Passenger vessels, on the other hand, remained at low levels despite a recovery during the summer months (Fig. 8, Supplementary Fig. 7).

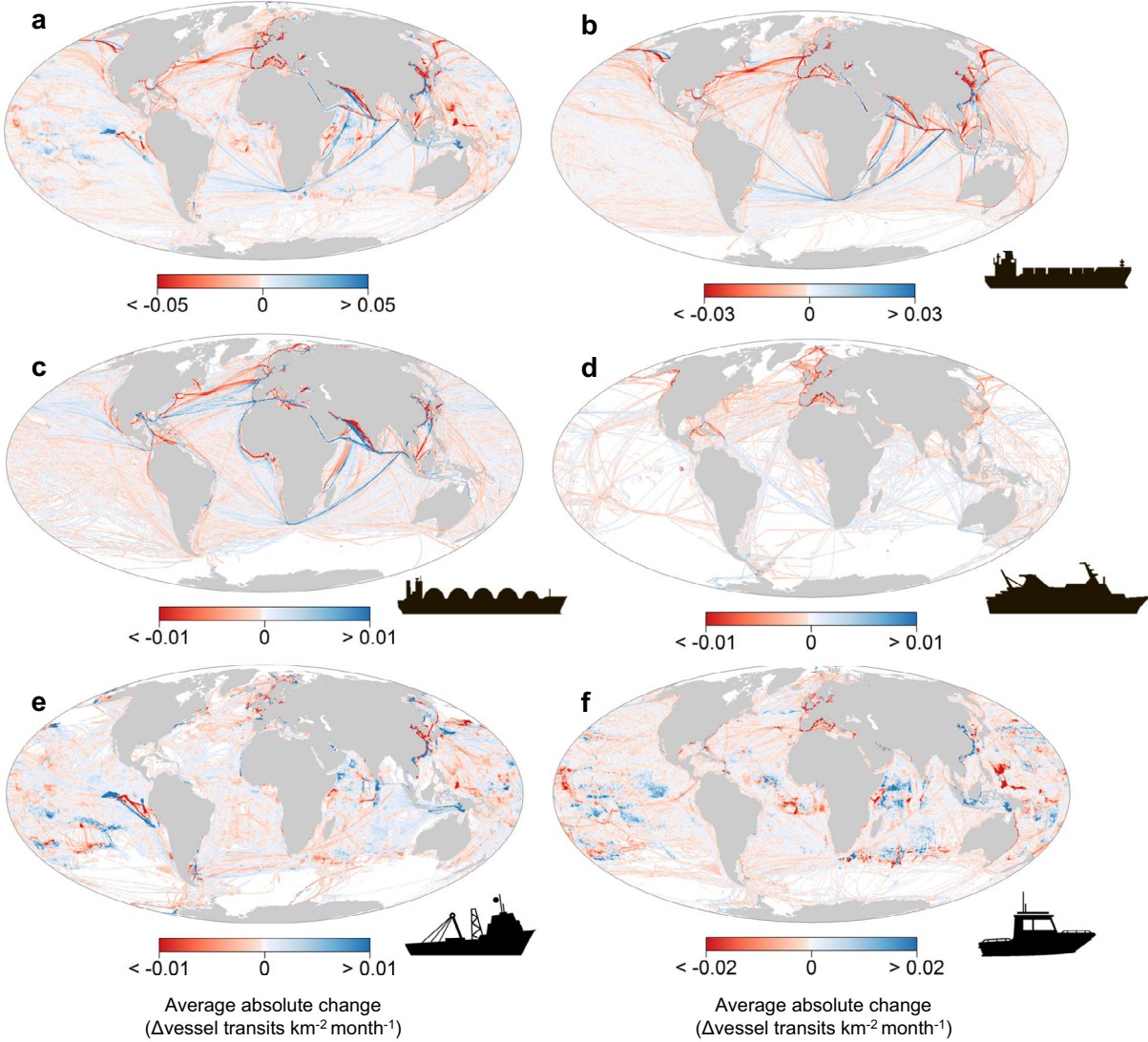

**Fig. 3 Global changes in vessel traffic density.** Maps showing the average of the absolute difference in traffic density (number of vessel transits per square kilometre per month) between equivalent months (January–June) from 2020 and the reference year 2019. Absolute differences derived using cell-by-cell subtraction. Negative (red) cells indicate an average reduction during 2020. Scale range values reflect the absolute maxima (99th percentile). Vessel categories: **a** all vessels, **b** cargo, **c** tanker, **d** passenger, **e** fishing and **f** other vessels.

## Discussion

Our oceans are responsible for the carriage of around 80% of world trade and are the lifeblood of many national economies which rely heavily on fishing and tourism[30,38,43,44]. Here, using AIS data we have quantified and mapped changes in ship-based activities to provide a comprehensive overview of how multiple lockdowns to counter COVID-19 have impacted maritime traffic. Our data-driven approach shows that this has led to an unprecedented impact at global and regional scales across all sectors—leading to a general decrease in vessel traffic, and variable changes in the operating behaviour of different sectors of transport, fishing and recreational vessels. This is the first time that it has been possible to monitor and map the response of shipping to such a sudden global disruption in near real-time.

At the global scale, our analyses reveal a decline in global marine traffic during the pandemic, a pattern mirrored across multiple maritime sectors at varying scales. European Seas, and in particular the Mediterranean Sea, were regions dominated by the greatest reductions in marine traffic highlighting the dramatic and rapid impact of lockdown measures on the movement of vessels in the northern hemisphere. East Asia, however, evidenced

a mixture of patterns and general increase of marine traffic particularly within China's EEZ, which likely reflects an upturn in economic activity associated with the general and earlier easing of lockdown measures relative to other countries which suffered outbreaks later. The rapid recovery of China´s activity has also been reported by other studies looking at $CO_2$ emissions[45,46].

Analyses such as ours provide an unparalleled opportunity to assess changes to the blue economy at global and regional scales. Most notably, our findings reveal that the COVID-19 outbreak has led to significant disruptions and regional slowdown in vessel activity that was sustained for several months along well established maritime transport routes and maritime chokepoints across Asia, Africa and Europe. However, the impact on the maritime transport sector (i.e. cargo vessels and tankers) was lower than in other sectors directly influenced by the lockdown measures and restrictions on travel, which reinforces global interconnectedness and reliance on many goods being manufactured overseas rather than locally. In contrast, the most heavily impacted and least resilient to COVID-19 were the tourism and recreation sectors, with major declines and slower recovery rates in vessel activity at global and regional scales. Such disruptions

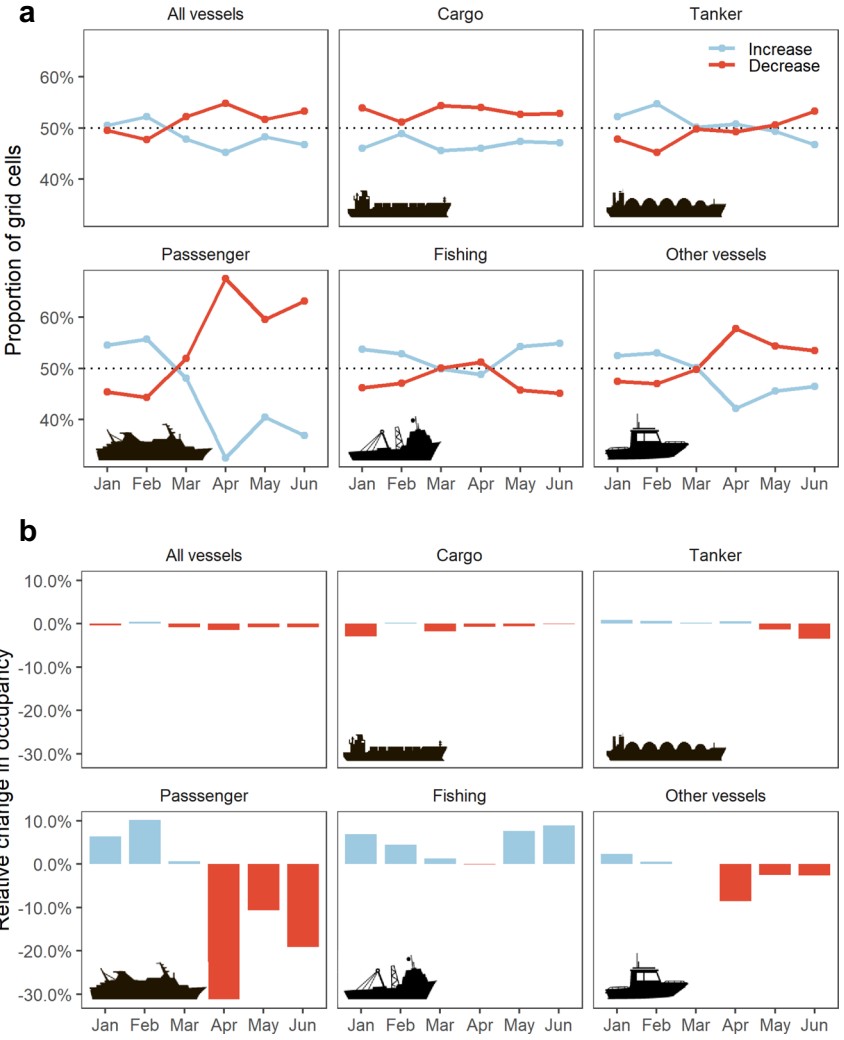

**Fig. 4 Global changes in marine traffic during COVID-19. a** Changes in traffic density represented as a proportion of grid cells with increases and decreases per month and vessel category. **b** Changes in occupancy estimates, reflecting gains and reductions of the extent used by each vessel category per month. All estimates in (**a**) and (**b**) are based on a global grid of 0.25 degrees resolution, comparing between equivalent months (January–June) from 2020 and the reference year 2019.

have the potential to turn into far-reaching and significant social and economic impacts on tourism-dependent economies for years to come.

For fisheries, the spatial heterogeneity of changes suggests that the impact of the outbreak has been uneven across fishing fleets. Regional analyses in the Western Mediterranean, however, reveal that fishing vessel activity was close to pre-lockdown levels by June, suggesting that the industrial fisheries sector, which is well-resourced and heavily subsidised in some countries[47], plays an important role in the global economy and so was resilient to COVID-19. Our work was not able to monitor changes in small-scale fisheries due to limitations of deployment of AIS in this sector, but previous studies suggest these fisheries, which dominate in many low-income countries, are likely to be particularly vulnerable to socioeconomic effects derived from COVID-19[10,48,49]. Further work is needed to ascertain the impact of the COVID-19 outbreak on the behaviour of small-scale fisheries sector.

Our global and regional assessments reveal spatial and temporal changes of ship-based activities in response to confinement measures. In addition, given the growing trends in marine traffic occupancy (estimated increase of ca. 3.0% in 2020, see Supplementary Methods), it is reasonable to assume that our

comparison with 2019 is providing a modest underestimate of the impacts of COVID-19. In fact, prior to the COVID-19 outbreak, there was a long-term acceleration of maritime activity in intensity and occupancy, including shipping and cruise tourism among others[39,43], with increasing rates of shipping in 92% of the EEZs[34], and forecast increases of the global shipping network ranging from 240% to 1209% by 2050 under different economic scenarios[20]. Conversely, not all changes observed were necessarily related to COVID-19. Changes in maritime activities can be driven by multiple factors such as regulations (e.g. marine protected areas, speed limits, traffic separation schemes), socioeconomic changes, piracy, environmental changes or by cultural and political events[29,30,50,51]. For example, the fall in oil price in early 2020 likely contributed to the observed increases of tanker vessel traffic before the declines in oil demand due to COVID-19[52,53] and the re-routing of cargo ships to avoid Suez Canal tolls in favour of the longer journey via the Cape of Good Hope[54]. On the other hand, the shape of displacements in fishing vessels intensity suggests several annual shifts in the fishing grounds (e.g. near Peru, Fig. 3e). Finally, other changes like the large increases of fishing vessels in Indonesia (Figs. 3e and 5d) could be attributed to a national regulation that enforced the use of AIS for all vessels by the 20th August 2019 (Ministerial Regulation PM

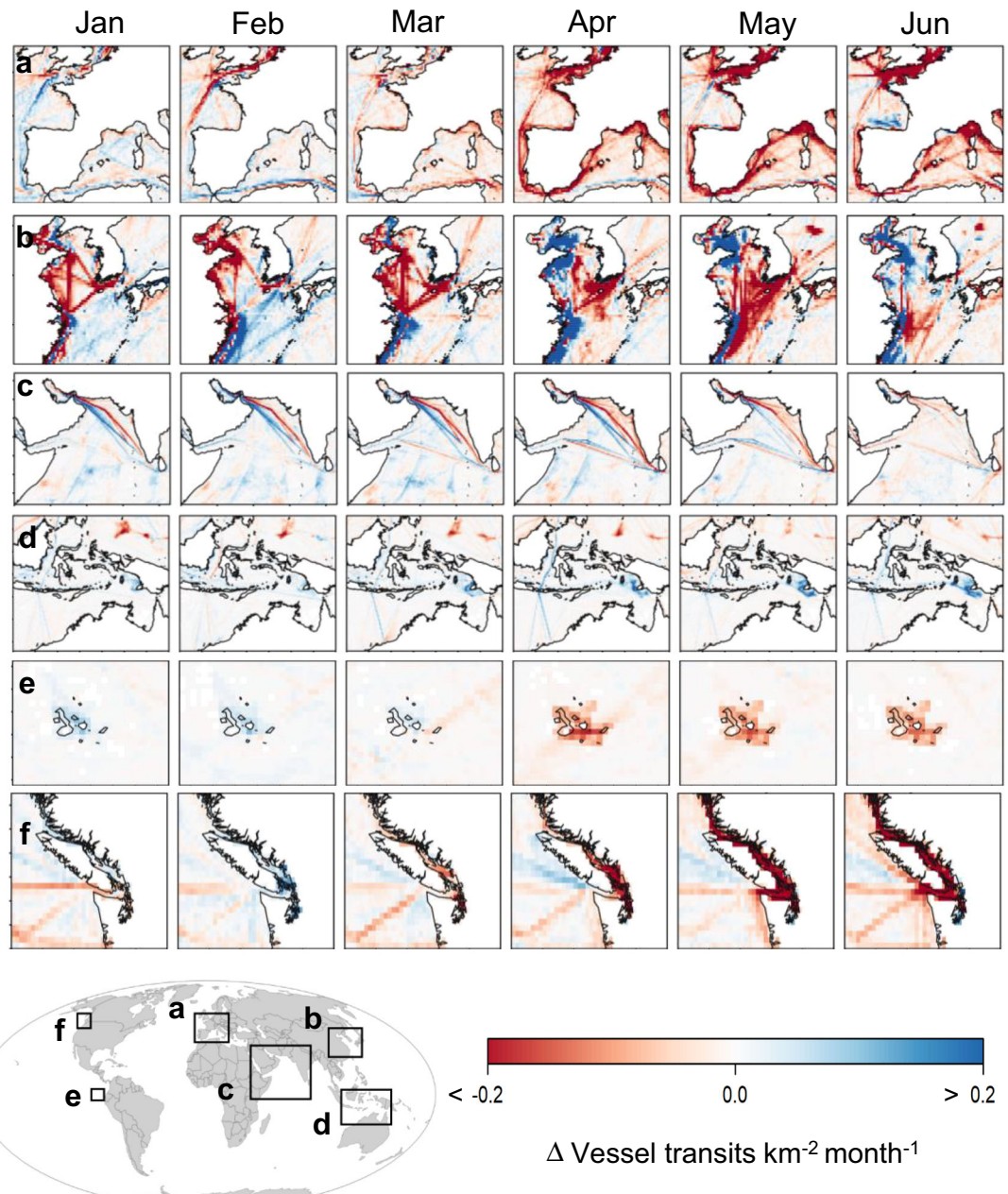

**Fig. 5 Regional and local monthly changes in vessel traffic density.** Maps showing the absolute difference in traffic density (number of vessel transits per square kilometre per month) between equivalent months (January–June) from 2020 and the reference year 2019. Absolute differences derived using cell-by-cell subtraction. Negative (red) cells indicate decreases during 2020. Focal areas include: **a** Western Europe, **b** East China Sea, **c** Arabian Sea, **d** Indonesia, **e** Galapagos Islands and **f** Port of Vancouver.

7/2019). Determining the fine detail of the degree to which specific observed changes were driven by COVID-19 or other factors will require further regional and local assessments.

Monitoring the movements of marine traffic in near real-time at a global scale is now possible as a result of unprecedented technological advances in the domains of big data and nano-satellite communication systems leading to increases in global AIS coverage. It is noteworthy that during the most recent comparable global shock, the 2008 financial crisis and associated recession, a study such as ours would not have been possible. In addition to gridded density maps, there are additional characteristics that can be derived from raw AIS data (e.g. port calls, individual vessel trajectories) that have already been used to assess possible transmission of COVID-19[13,36,37]. Furthermore,

changes in the properties of the global shipping network are essential to better understand the effects of COVID-19 on world trade or assess changes in the derived risk of biological invasions[20,38]. Moreover, using trajectory information to quantify changes in vessel behaviour would allow mapping changes of multiple human pressures (e.g. underwater noise, fishing effort, boat anchoring, air pollution), to assess their interactions and potential effects on wildlife[1,35] and quantify their cumulative impacts on marine ecosystems[34,55]. Several ongoing initiatives providing emerging AIS-derived products at multiple spatial and temporal scales (e.g. EMODnet Human activities, and UN Global Platform AIS Task Team Initiative) will prove essential to allow large-scale monitoring of the progress and potential effects of COVID-19 and other future shocks.

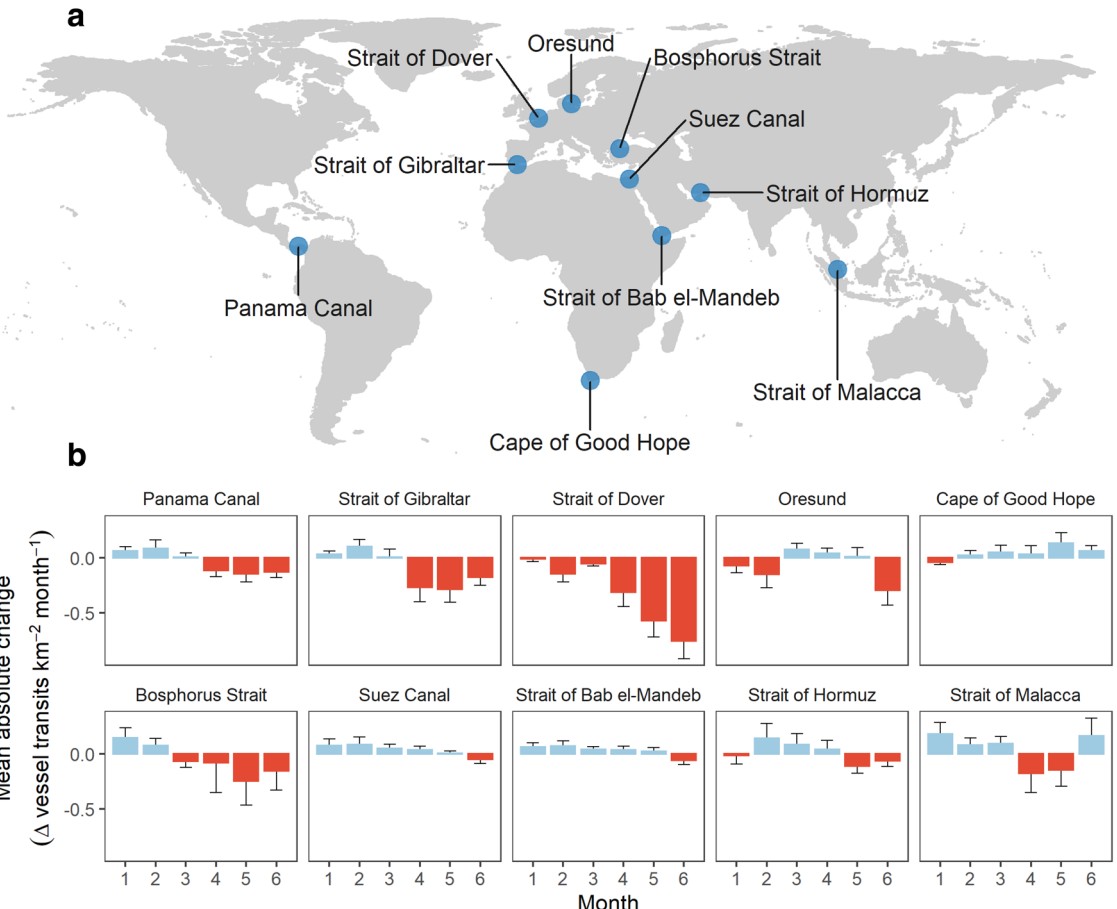

**Fig. 6 Changes in marine traffic density in maritime chokepoints. a** Map showing the location of maritime chokepoints ($n = 10$). **b** Mean absolute change of marine traffic per month per chokepoint estimated within a 0.5 degrees radius from each chokepoint. Error bars represent standard deviation. Change estimates in comparison with monthly densities from equivalent month from reference year 2019, with increases coloured in blue and decreases in red. Months numbered January (1)–June (6).

Unintended short-term cessation of human mobility due to natural catastrophes, electrical blackouts or terrorist attacks resulted in previous "natural experiments" that unmasked the effects of human activities on the Earth System[8]. For example, the reduction of air traffic as a consequence of 11 September 2001 terrorist attacks in the USA offered the possibility to discern the effect of condensation trails from jet aircraft on daily temperature ranges[56]. While the COVID-19 pandemic has brought a dramatic global health and socioeconomic crisis, the unprecedented disruption during lockdowns offers new opportunities for environmental research[1,8]. For instance, substantial decrease in noise resulting from confinement measures offered the chance to extract anthropogenic sources of noise from those of natural processes[57] and assess responses of birds to recently vacated acoustic spaces[58]. The spatial and temporal heterogeneity of changes described in this study is highly relevant for further studies aiming to assess the environmental effects of COVID-19 on marine ecosystems. Commercial fishing and shipping, in fact, contribute significantly to overall cumulative human impacts on the ocean[55] and information about their spatial patterns is of paramount importance for conservation planning[59,60]. The reduction of maritime activities in some affected regions and locations may provide some positive outcomes for the marine environment. For instance, after the 2008–2009 global financial crisis, decreases in bottom trawling fishing pressure and reductions in vessels speeds (i.e. due to increase of fuel price[51]), resulted in improvements in benthic ecological status[14] and air

quality[15], respectively. Our global assessment is well aligned in time with two focal studies that reported reductions of marine traffic in the Port of Vancouver and Venice during earlier stages of the COVID-19 pandemic, documenting declines in levels of underwater sound[23] and water turbidity[24]. Such agreement highlights the potential of our global dataset to identify impacted and control locations for comparison in further environmental studies. In addition, reductions found in vessel traffic around the Galapagos Islands or coastal areas of the Mediterranean also suggest that marine protected areas could benefit from a decrease in marine traffic associated with tourism and other human-derived pressures. Conversely, if an associated reduction of surveillance effort by maritime authorities occurs, this could present a higher risk of illicit activities (e.g. illegal fishing, trafficking of drugs), especially in lower-income countries[48,61]. In fact, our results suggest there were increases in industrial fishing activity in the national waters of some countries, a pattern worthy of further investigation; especially considering that illegal fishing or non-compliance with fisheries laws may have increased as a result of perceived or real reductions in enforcement efforts due to reduced logistical, personnel and financial resources during the COVID-19 outbreak.

Changes in marine traffic have been shaped by policy actions related to COVID-19 restrictions on human mobility and perturbations in consumer demand and supply chains. The response of marine ecosystems to COVID-19 will depend on the intensity and duration of the reduction of human pressures. There is,

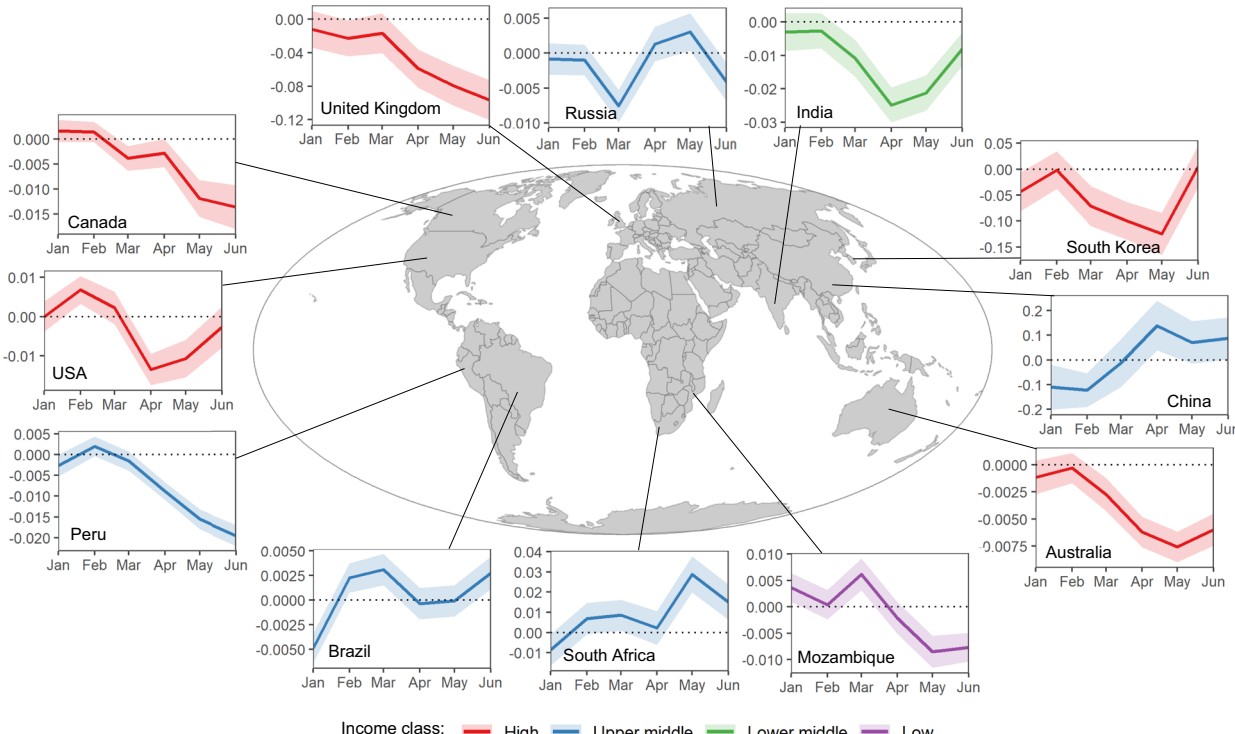

**Fig. 7 Changes in marine traffic density in selected countries.** Mean absolute change and standard deviation of marine traffic (number of vessel transits per square kilometre per month) estimated within country Economic Exclusive Zones (EEZ). Colour categories represent income classes from the World Bank. Change estimated in comparison with monthly densities from equivalent month from reference year 2019.

however, a degree of uncertainty around future scenarios and long-lasting impacts. The scientific community needs empirical observations in order to better understand the socioeconomic impacts on maritime sectors and the environmental consequences of COVID-19 on marine ecosystems. The pandemic has also constrained the capacities of research institutions to pursue monitoring programmes (e.g. research cruises) underscoring the need to advance implementation of real-time autonomous monitoring systems to survey the ocean, including anthropogenic impacts. As we continue to update changes in marine traffic density, we will be able to track longer-term changes due to COVID-19. Future AIS studies should address temporal variability of spatial patterns of more regional and sectoral focused studies. Such assessments will provide crucial insights into the effects of the current pandemic, or other global shocks, on the blue economy and ocean health.

## Methods

**Satellite AIS data**. For global analyses, satellite AIS (S-AIS) data for January–June in both 2019 and 2020 were obtained from exactEarth Ltd (http://www.exactearth.com/), a space-based data service provider which operates a constellation of 65 microsatellites to provide global AIS coverage at a high-frequency rate (<5 min average update rate). The latest upgrade in the constellation entered into production in February 2019 with the addition of 6 new satellites (i.e. there were 59 satellites in January 2019), thus S-AIS coverage can be considered equivalent for the study period (exactEarth Ltd. pers comm.). Values represented the monthly number of vessel transits within grid cells of 0.25 × 0.25 degrees. Vessels were classified into five categories: cargo, tanker, passenger, fishing and "other". The category "other" included any vessel not covered by the preceding categories (e.g. vessels conducting surveys and logistical services for industry, research vessels, recreational boats). We calculated the vessel density as the number of vessel transits per unit area, considering the difference of cell size across the latitudinal gradient[28]. Further details regarding post-processing and quality control procedures for density maps can be found in the Supplementary Methods.

**Terrestrial AIS data**. In addition to satellite stations, AIS signals can also be detected by terrestrial antennas (T-AIS). Unlike S-AIS, with global coverage, land-based antennas have a horizontal range of about 40 nautical miles (74 km).

Terrestrial AIS (T-AIS) data from the Western Mediterranean (map inset Fig. 8a) were collated by the Balearic Islands Coastal Observing and Forecasting System (SOCIB[62]) using a real-time operational system connected to a web-service provided by Marine Traffic (https://www.marinetraffic.com/). The database used in this study contained AIS data from 1 January 2016 until 30 November 2020 at 5-min intervals (comprising > 545 million AIS messages). In addition to the vessel tracks, the database also included information associated with each vessel, such as the vessel type or length. A first pre-processing of the raw data included the removal of duplicates, invalid identification numbers (i.e. Maritime Mobile Service Identity -MMSI- codes without 9 digits) and codes outside the correct numerical range (i.e. MMSI codes with first digits between 2 and 7 are those intended for individual ships). In order to address inconsistencies in the vessel and MMSI combinations (e.g., changes of MMSI across years), we selected the more frequent combination of MMSI and vessel characteristics (e.g. vessel name and vessel type) for each calendar year. We used a similar vessel categorization as the S-AIS dataset, but were able to derive a sixth category from the AIS metadata, separating "recreational" boats from "other" vessels. Therefore, vessels were classified into six categories: cargo, tanker, passenger (included high-speed craft and passenger vessels), fishing, recreational (included sailing vessels and pleasure craft; vessel type codes 36 and 37), and others (all other vessel types). We excluded ship type codes 20–29 (i.e. wing-in-ground-effect and search and rescue aircraft), as well as codes that had an invalid value (i.e. empty or null) or those where the value was not listed in the previous type codes. We calculated the number of vessels per day considering only those that were underway, thus removing moored vessels inside ports that were inactive. T-AIS coverage was not homogenous in the study area due to lack of uniformity in the distribution of antennas (see Supplementary Methods). Consequently, we filtered data to include vessels within the coastal zone (44.4 km, ~24 nautical miles) of EU countries (i.e. a total area of 164,318.2 km² comprised by Spain, France and Italy), thus reducing potential bias due to spatial and temporal gaps in signal reception.

**Changes in response to COVID-19**. In order to quantify changes in response to COVID-19, we compared traffic density estimates in 2020 with the same reference period in 2019. Recent studies have used similar approaches to assess the environmental impacts of COVID-19 (e.g. on air pollution[45,46]). In our study, this approach is consistent with S-AIS coverage (see above) and allows accounting for seasonal variability (Supplementary Fig. 6). Moreover, longer-term datasets do not suggest the presence of anomalies in 2019 (Supplementary Fig. 6, Supplementary Fig. 9). At a global level, we calculated the absolute and percentage change in traffic density and occupancy from baseline on a grid cell and monthly basis. We summarized density changes as function of latitude and distance to the coastline

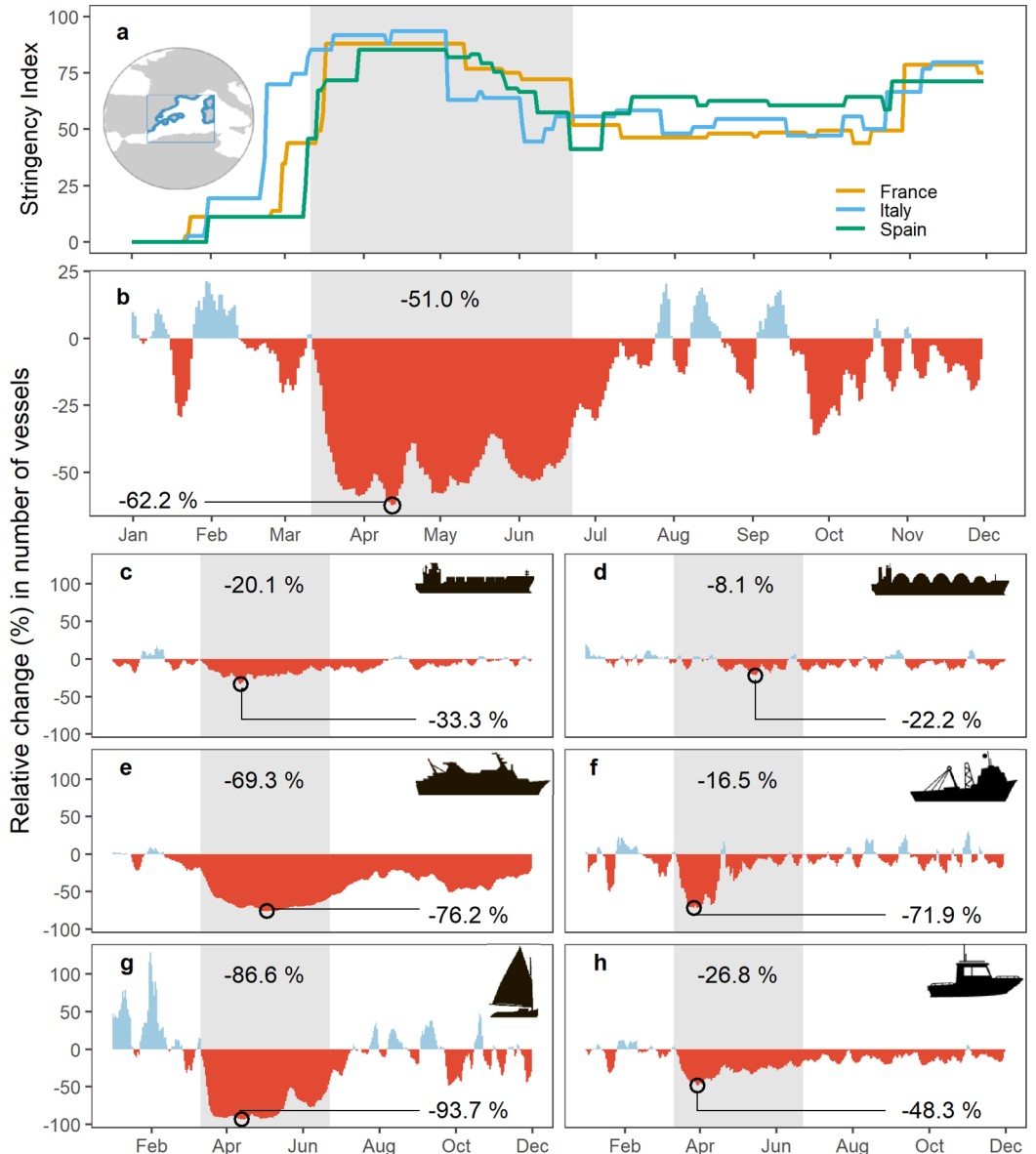

**Fig. 8 Daily relative change in number of vessels equipped with Automatic Identification System (AIS) transponders between 2020 and 2019 in the Western Mediterranean. a** Daily changes in the Stringency Index (100 = strictest response) as an indicator of confinement measures for EU countries present in the study area (i.e., Spain, France, Italy). **b–h** Relative change of daily number of vessels underway within the coastal zone (24 nautical miles) present in the study area per vessel category: **b** All vessel types, **c** cargo, **d** tanker, **e** passenger, **f** fishing, **g** recreational and **h** others. Increases and decreases in number of vessels are coloured in blue and red, respectively. Daily estimates using 7-day moving average (from 1st January to 30th November). The shaded area in grey highlights the period between the World Health Organization pandemic declaration on 11 March 2020 and 22 June 2020, when EU countries relaxed their confinement measures after the first wave. Values within the shaded area represent the median relative change during that period. Circles indicate maximal changes. Blue area in the map inset on part (**a**) represents the spatial extent of the regional AIS dataset.

using zonal statistics (i.e. average and standard deviation at regular intervals). Moreover, we conducted an estimation of expected occupancy for 2020 based on the last 10 years of ship density estimates from altimetry data[50] (See Supplementary Methods). In addition, we estimated density changes at 10 maritime chokepoints[41,42]. For each chokepoint, we defined a buffer of 0.5 degrees and estimated their mean and standard deviation of the absolute change in traffic density on a monthly basis.

At a regional level (i.e. Western Mediterranean), we compared the unique number of vessels on a daily basis using T-AIS. Our dataset showed a marked annual cycle, reducing in the boreal winter and a year on year increasing annual trend for some sectors (Supplementary Fig. 6), hence we compared the 2020 values (since 1st January to account for changes from early in the development of the pandemic) with the same periods of 2019. In order to take into account the dynamics of ship-based activities through time, the comparison between the datasets of the 2 years was adjusted so the same days of the week were being compared and to allow for the extra day in 2020, being a leap year. We calculated a 7-day moving average and then estimated the relative percentage change.

**Effect of containment measures**. We evaluated the effect of containment measures on changes in marine traffic density at country level. Monthly gridded density values were averaged by Exclusive Economic Zones (EEZ)[63] to calculate the percentage change in traffic density in 2020 from baseline (i.e. 2019). In order to have the best possible alignment with the containment measures (i.e. provided at country level), we selected EEZs pertaining to coastal countries, hence excluding territorial regions (e.g. overseas territories), EEZs without traffic density data (e.g. those within the Caspian Sea), joint regimes (e.g. shared and jointly managed EEZs), as well as disputed or unclaimed areas. Furthermore, we selected regions larger than 2307 km$^2$ (i.e. equivalent to the size of three grid cells in the Equator) to ensure we had at least three observations per EEZ. Then, we obtained information on income levels from the World Bank and containment measures from the Oxford COVID-19 Government Response Tracker (OxCGRT) for each of the selected countries ($n = 124$). OXCGRT provides a transparent, real-time monitoring system that allows comparison of government measures between countries[40]. We estimated the monthly median Stringency index (Index methodology version 3.1). This index is an additive score of nine policy decision

indicators, rescaled to vary from 0 to 100, which records the strictness of the lockdown measures per country. To analyse the effect of confinement measures on the relative change of vessel traffic density we used linear mixed-effects models (LMM). Mixed-effect models are a useful tool in presence of repeated measurements for units of observation that are clustered (e.g. within geographic regions)[64]. LMMs, for instance, have been used previously to assess temporal changes in ocean health on EEZs[65] or the effects of air pollution on severity of COVID-19[66]. We included the interaction of Stringency index and income levels as fixed effects, with country as random effect. We ran a separate model for each ship category. The total number of sampling units (i.e. countries) used in the models varied due to differences in occupancy between categories (e.g. passenger vessels had lower occupancy than cargo vessels; hence model is based on lower number of countries, Supplementary Table 1). LMM's were run in the program R[67] using the restricted maximum likelihood (REML) estimation within the lme4 package[68]. Significance was calculated using the lmerTest package[69], which applies Satterthwaite's method to estimate degrees of freedom and generate $p$-values for mixed models.

## Data availability
Stringency index data are available from the Oxford COVID-19 Government Response Tracker (www.bsg.ox.ac.uk/covidtracker). Raw AIS data are available from SOCIB and exactEarth Ltd. Anonymized and aggregated data from terrestrial AIS are available (https://doi.org/10.6084/m9.figshare.12667256). Density maps of satellite AIS were purchased from exactEarth Ltd., are used under license, and cannot be publicly shared by the authors. We make the global difference maps publicly available (https://doi.org/10.6084/m9.figshare.12676070).

## Code availability
All analyses were coded in R version 3.6.0[67]. Code which is available from Github at https://github.com/dmarch/covid19-ais and https://doi.org/10.5281/zenodo.4582712.

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

## Acknowledgements

We thank Anastassis Touros from Marine Traffic and Simon Chesworth from exactEarth Ltd. for support with AIS data. We thank Belén Sáez for her contribution during various stages of the paper's preparation. D.M. and J.T. acknowledge support from the European Union's Horizon 2020 research and innovation programme (Marie Skłodowska-Curie grant agreement no. 794938, EuroSea grant agreement no. 862626 and JERICO-S3 grant agreement no. 871153). B.J.G. is supported by NERC Grant NE/V009354/1. K.M. is supported by the Waterloo Foundation, and the Darwin Initiative (Project 26-014) through funding from the Department for Environment, Food & Rural Affairs (Defra) in the UK.

## Author contributions

D.M. and B.J.G. conceived and designed the study. D.M. performed the analysis. D.M. and B.J.G. wrote the manuscript with input from J.T. and K.M. All authors approved the final version of the manuscript.

## Competing interests

The authors declare no competing interests.
