## [Peer Review File · Nature Communications]

REVIEWER COMMENTS

Reviewer #1 (Remarks to the Author):

The authors provide an interesting analysis of the effects of Covid 19 on different types of marine traffic. There are no great concerns with the methods employed by the authors, and while I have a few concerns and comments in presentation these should be relatively easy to address. The readability of the paper was challenged by reference to a large number of supplementary tables and figures, referenced at regular intervals. In many cases the supplementary figures come across as integral rather than supplementary. Overall however I appreciate this data being communicated rather than omitted. In many ways the paper would be more interesting if run for a longer timeframe to better chart recovery (or not) from Covid-19 impacts.

Greater care should be taken with reference to log percentage change, as well as percentage in its standard sense. The broad readership would benefit from greater explanation of the L% metric, and guidance in how to interpret it. What does -276 L% equate to? Where possible I recommend showing absolute and L% decline, as what doesn't come across easily are the magnitude of changes and some transference into more common language terms would also help 'more than doubled', 'tripled', 'halved' etc. Where figures and tables reflect L% this should be quoted on both the axes/column headers and legends of the figures. I would suggest sticking to the absolute numbers (as has been done with the main figure set) or more conventionally understood metrics.

It is disappointing that the global analysis was restricted to April, while the justification for choosing this month is well communicated, expanding the analysis to the preceding and subsequent months would have added greater depth to the analysis and possibly identified lags and different regional phases of impacts.

My more specific concerns are addressed by line below:

-L19: 'which sectors are most affected, in which regions, and for how long' I am concerned by the for how long part of this statement. Has the study been run sufficiently post covid to capture the 'how long' part of this statement?

-L56: should use caps on Automatic Identification System?

-L69: Western Mediterranean Sea. The location seems to be driven by access to high level AIS data, if possible expansion to the whole Mediterranean would be appreciated to capture movement of traffic into and out as well as within.

-L71: 'most impacted', I suggest use term 'heavily impacted'. The outbreak is still underway and is increasing in some European countries. How do you quantify and qualify 'impact' in this case? Number of cases, deaths, GDP, etc. all of these are sensitive to country size as well as a host of other factors?

-L84 (also L98, L107 etc): "Global marine traffic in April 2020 was present in nearly 76.9% of the ocean". Such statistics are picked up and quoted by media without understanding the full meaning and such high percentages are only achieved because of the relatively large grid area used in analysis. This leads to a fallacy when comparing the footprint of a vessel track (the width of a vessel or its operations, along the length of its track) to much larger grid cell. See detailed comments of Amoroso et al. in Science 2018 on global estimates of fishing supplied in Kroodsma et al., 2018. Use of 0.25 deg grid data in the present study fits within the range of areas estimated. Lower end of scale used in fig 2a, 0.001 vessels per km² implies most of cell is empty of traffic, and 80% cells have a density of < 0.0042. The above statement is therefore misleading, especially when moved out of context. <https://science.sciencemag.org/content/361/6404/eaat6713/tab-pdf>. "We accessed the 0.01° grid fishing data made available by Global Fishing Watch (2) and reanalyzed these data at resolutions of ~3100, ~123, and ~1.23 km² (corresponding to 0.5°, 0.1°, and 0.01° at the equator), giving footprint estimates of 49%, 27%, and 4% of ocean area, respectively. Thus, higher-resolution analyses reduced their global fishing footprint estimates by a factor of >10." Use of absolute numbers is more accurate.

- L99 / L103 Global average decrease of 3.3%, 9.4% relative decrease. Is this L%. If not, why use L% elsewhere and not here. If so, make clear it is L%.
- L174 "Electronic Vessel Monitoring Systems" Up to now we have focussed on AIS, a vessel monitoring system (VMS) has a specific meaning in a fisheries context, suggest keep as AIS
- L175 China's EEZ. Happy to be corrected, but my understanding is China never instigated a national lockdown. The majority of china which represents ~19% global GDP remained open.
- L197: '240-1,209% by 2050' unclear what is meant by this. Typo?
- L201: regional shutdown And Europe. There appears to be an increase in traffic out of Gulf of Aden. Is this coming through from the med via Suez? Some focus on impacts through other trade "pinch points" such as Suez and the Panama canal, and well as Strait of Malacca would also be appreciated.
- L214: 'is less likely to be affected than the more vulnerable small-scale fisheries sector that dominates fisheries in many lower-income countries' The authors should note at this stage that they are unable to monitor these particular fisheries through AIS (it is not referenced until L240). It is mentioned later but relevant here... e.g. this study was not able to assess impacts to small scale fisheries due to limitations of AIS, but studies elsewhere suggest these fisheries are likely to be particularly vulnerable.
- L229 'a recent regulation....' Can a reference be provided for this, or more detail on what the regulation mandates in terms of AIS?
- L240 limitations of AIS data – it would be helpful if authors provided this information at the outset of results and discussion.
- L242: 'In this study, we used gridded density maps at the finest data resolution (0.25 degrees) available at global scale and provided on an operational basis'. See previous points on how % and vessel density is presented. Again it would be helpful to present this information earlier.
- L244: 'There are additional characteristics that could be derived from raw AIS data (e.g. port calls, individual vessel trajectories) that warrant further attention' You could reference this statement from references already in your list to show that this has can already be done. What you are specifically suggesting is that this is done with respect to COvid 19 e.g. 22, 23 etc.
- L253: 'there is, as yet, no international body providing open access to shipping tracking data at a global level'. While we would all like such data to be free, for me this comment misses the advantages that commercial providers have taken the risk of putting constellations of receivers into space at considerable expense and risk and must therefore operate a viable commercial offering to maintain the data service. For the data to be provided for free would therefore require an international body to adequately recompense the providers. Can the authors make a more compelling commercial argument on the cost/benefit of providing such data free as a service to the international community?
- L262 'Previous economic crises....' Sentence reads very long. Possibly a missing semi colon, or split into two.
- L268 'congruent with' I suggest something more specific, such as 'aligned in time with two focal studies'.
- L269 'resulting in' suggest reword to something such as that document
- L270 'respectively' not needed as reference goes with example.
- L272 'our results...could benefit from reduction in marine traffic.' I fail to see how this linkage to MPAs has been shown in the results?
- L273: unclear where this comment on reduction of surveillance effort comes from. What types of surveillance.
- L289: A previous comment suggested the need for more regional and sectoral focused studies, while this one calls for a more global approach. For me this paper does a good broad brush job of capturing the global but will have inevitably missed more local or focussed impacts. I suggest the focus should be more on the former, or you should be more specific about the types of global study and integrated thinking that are needed.
- L363. Greater background should be provided on the logarithmic percentage change metric beyond citation. As discussed above I think this is perhaps needed more for the results and interpretation than here in the methods.
- L368 ABNJ. Occasionally in tables/figures use the term High Seas. Correct, but perhaps confusing and

unnecessary. I suggest you use a single term ABNJ.

-L380: 2019 reference. Is there any effort to assess general variability around the 2019 reference point? E.g. How variable was 2019 compared to 2018, 2017 etc. How does the 2020 period compare to background variance? Just looking at the pre-covid 2020 data there remain some periods for some vessel types that were quite different (granted not as large).

FIGURES & TABLES:

-Fig.3. ABNJ should read ABNJ sub-regions

-Table S1 – see previous comments on concern calculating % area based on a given grid cell. Occupancy columns therefore concern me.

-Table S2: Standard date format of dd-mm-yyyy preferred.

-Fig. S2 Here and several similar figures. No units on color bar or in legend. Is colour scale linear-presumably logarithmic as in Fig 2a? It would be helpful to have some additional ticks, and scale values.

-Fig. S3 It would be helpful to have some additional ticks, and scale values.

-Fig S4: y axis should show L% not %

-Fig. S5 x axis should show L% not %

-Fig S7 should read ABNJ Sub regions

-Fig. S11 L% on y axis, ROI on inset globe difficult to read

-Supplementary Data tables 1-4 replace column header with L% rather than %.

Reviewer #2 (Remarks to the Author):

This paper does an excellent job quantifying changes in vessel activity related to COVID-19 lockdown measures. I appreciated seeing these values displayed in space, time and by ship type, as well as zoomed into the Mediterranean for more detailed analyses. The Stringency index is quite useful as a metric for contrasting ship movements to the intensity of governmental mitigation measures to COVID-19, such as lockdowns. This paper is particularly relevant to the marine ecological and blue economy fields as a baseline for subsequent study on how shipping may impact ecological status and other economic activities in the context of assessment before, during and after COVID-19 as a natural experiment. Towards this aim, I am especially pleased to see the free availability of the spatial and tabular datasets along with the R code for conducting the analysis and visualization.

I recommend this manuscript for publication as it lucidly presents its novel findings unique to our COVID times as well as relevance for further study across the marine scientific community. Nonetheless I would like to share some additional minor suggestions for improvement.

Wording:

Replace "ABJN" (lines 368, 588) with "ABNJ" (Areas Beyond National Jurisdiction)

Replace "vessel monitoring systems" (line 175) intended generically here with another term so as to not confuse the AIS data used in this study with that from a separate and different ship data type called "vessel monitoring system" (VMS) data (e.g. see <https://globalfishingwatch.org/vessel-tracking-data/>)

The phrase "entered into force" (line 229) seems odd, so could be replaced with "became enforced". Although described in the Methods, some mention in the Introduction of the shift from the AIS receivers being on land to satellite would help clarify how global monitoring is technically feasible in the sentence at lines 56-7.

Similarly in the Introduction, I would like to see explicit mention of how AIS is generally limited to bigger ships (e.g. > 300 tons) that is well described in the Methods.

Methods:

Can you produce a direct correlation between Stringency Index and shipping activity by country? This would presumably be done spatially within a country's surrounding EEZ waters (versus the nationality

of ship origin). This could be assessed at various time lags. This would then provide a coefficient (for instance from a linear model) predicting the amount of shipping reduction resulting from the level of Stringency with a measure of uncertainty.

The comparison of global marine traffic between April of 2019 and 2020 is quantitatively illustrative of the reduction (1.4% global occupancy), but I am also curious to know how April 2020 compares with the expectation of April 2020 marine traffic without COVID if it continued with its expected growing trend -- a presumably bigger difference.

I appreciate the finer scale Mediterranean analysis beyond the global. This also begs the question of how much more information are you getting from using the terrestrial versus satellite data? Is there a simple comparison between the two over the same area that you could make? For instance, the denser the number of AIS signals in busy shipping areas, the higher the dropout rate for the satellite receiver. Mention of these shortcomings in the global dataset justifies this more detailed view.

Discussion:

Although some relevant examples in marine traffic were mentioned, it might be interesting to note how other 'natural experiments' have influenced our understanding of ecological impacts and societal decision-making. For instance, how did 9/11 and the momentary cessation of plane traffic influence our ecological understanding of planes on birds?

Reviewer #3 (Remarks to the Author):

Summary

The manuscript uses AIS tracking data from ships to characterise changes in the density of global shipping at varying spatial scales. The Stringency Index was used to indicate when lockdown measures were at their highest, in April 2020. AIS data were sourced from satellites (S-AIS) and terrestrial receivers (T-AIS). S-AIS data were analysed in April 2019 (as a comparator benchmark) and April 2020, and the results were compared through cell-by-cell subtraction at different spatial scales of grid cell, EEZ, ABJN, marine ecoregions, and FAO major fishing areas. T-AIS data from the Mediterranean Sea were analysed from 2016-2020 to show how the number of vessels changed over time. The global results give a snapshot in time of how vessel density changed at different spatial scales.

General

The writing and figures in the manuscript are of high quality, and clearly a lot of time and thought has been put into producing the set of images, which are easily interpretable and well presented.

However, I do not think the manuscript, as it is now, contributes to the scientific knowledge base sufficiently enough to warrant publication. Set out below are the key areas that the manuscript needs to address to become publishable by Nature Communications.

1. Stringency Index – this was in a Europe-centric context. The authors use the rationale that the majority of Europe had the most stringent lockdown measures in place in April 2020, which seems plausible given Fig. 1. However, global maps of vessel density are then presented at this time point, at a time when many countries that have a high number of vessels associated with them (e.g. China, Norway, Singapore, Norway, USA, S. Korea, Denmark), and/or have significant fishing industries (e.g. China, Indonesia, USA, Philippines, Norway, S. Korea), were not in the most stringent lockdown. The Europe-centric view not only limits the interpretation of the global maps that have been produced, but undermines their validity because some of the interpretation is discussed in relation to areas that were not in full lockdown at the time (Fig 2d, e, f, h). This must be accounted for in the analysis. It is reasonable to assume that countries that were in lockdown would have reduced vessel traffic and countries that were not in lockdown or in a less severe lockdown would have increased or similar traffic. What this analysis does not do is show how this trend plays out when taking into account the underlying restrictions over time.

2. The analysis of cell-by-cell subtraction does not take account of dependency between the grid cells. Vessel densities are bound to be spatially correlated in that grid cells near each other are more similar. Without accounting for this, a cell-by-cell subtraction is likely to be biased. How do you know whether a change is significant? There is literature on comparing spatial differences such as Hagen-Zanker (2006), Moran's I or Geary's C tests (Cliff and Ord (1970), or Environmental Niche Models (Warren et al, 2008).

3. Limitations – there are caveats and limitations around using AIS data, and some of these are mentioned (Line 240). However, there are a number of other caveats, for example, locations from fishing vessels may not be as accurate as other vessel types because AIS on these vessels can be activated manually. The manuscript mentions few antennas for T-AIS in north Africa (Line 355) and temporal bias was reduced but was there also spatial bias due to non-homogeneity of antenna, and if so, how can this be accounted for?

Specific points

Line 26 – Guidance of 'potential effects of COVID-19' have not been demonstrated in this manuscript.

Line 73 - Although socio-economic and environmental effects of COVID-19 are mentioned, it is not clear how measuring changes in vessel density can provide guidance on these effects? How can this information be linked explicitly?

Line 150 – This is only the case because of the time point (April 2020) chosen for the analysis. If the time point was earlier in the year when China was locked down or later when the US / South America, and Africa had more severe restrictions, shipping activities, and therefore change in density, would look very different.

Line 184 – This is solely because shipping traffic aggregates close to the coast.

Line 186 – You must caveat this because this is partly due to the choice of time point (April 2020) and you found the Mediterranean Sea is particularly affected because of the fine-scale analysis you undertook.

Lines 202-203 – This is the first time the Silk Road and Strait of Malacca has been mentioned – where is this analysis?

Line 205 – Where is the analysis that shows an increase in demand for oil tankers? Fig. S11 shows smaller increases and larger decreases.

Line 289 – Rerunning the analysis from different time points would provide different snapshots but how could that then be used to give facilitate insight into long-term impacts, and how does demonstrating changes in vessel density relate to anthropogenic impacts on the marine environment?

Line 329 – What is the rationale for choosing <100 cells? Is this an arbitrary threshold?

Line 330 – Remove the word 'accurate' or rephrase the sentence. Any projection is accurate for some property, in this case it's area.

Line 345 – How did you define the additional category of 'recreational'?

Line 369 – 'We averaged per-pixel values, allowing direct comparison among regions despite large differences in size' – it would have been more informative to use the standard deviation from those mean values in the comparison.

Line 371 – This could bias your results. You need to explain how you included them in the analysis if the spatial scales were not the same. The rationale for including the EEZs is questionable – firstly, this analysis is not about air pollution or underwater noise and it doesn't link to it; secondly stating that environmental pressures are 'diffuse' isn't a rationale.

Authors' response to Decision Letter on NCOMMS-20-28656
"Tracking the global reduction of marine traffic during the COVID-19 pandemic "

RESPONSE TO REVIEWER COMMENTS

Reviewer #1:

The authors provide an interesting analysis of the effects of Covid 19 on different types of marine traffic. There are no great concerns with the methods employed by the authors, and while I have a few concerns and comments in presentation these should be relatively easy to address. The readability of the paper was challenged by reference to a large number of supplementary tables and figures, referenced at regular intervals. In many cases the supplementary figures come across as integral rather than supplementary. Overall however I appreciate this data being communicated rather than omitted. In many ways the paper would be more interesting if run for a longer timeframe to better chart recovery (or not) from Covid-19 impacts.

Response: *We acknowledge these helpful suggestions pointed out by the reviewer. We have worked to improve readability of the manuscript, by: (1) moving some key figures into the main manuscript and (2) minimizing the number of references to the Supporting Information. Moreover, we have extended both global (January-June) and regional datasets (updated until 30th November 2020), thus providing a longer timeframe.*

Greater care should be taken with reference to log percentage change, as well as percentage in its standard sense. The broad readership would benefit from greater explanation of the L% metric, and guidance in how to interpret it. What does -276 L% equate to? Where possible I recommend showing absolute and L% decline, as what doesn't come across easily are the magnitude of changes and some transference into more common language terms would also help 'more than doubled', 'tripled', 'halved' etc. Where figures and tables reflect L% this should be quoted on both the axes/column headers and legends of the figures. I would suggest sticking to the absolute numbers (as has been done with the main figure set) or more conventionally understood metrics.

Response: *We acknowledge the selection of the metric to represent relative change can bring confusion to the reader. We have replaced the Log change (%L) by the conventional relative change (i.e. $X_{2020} - X_{2019} / X_{2019}$). Whenever appropriate, we have provided both absolute and relative change numbers.*

It is disappointing that the global analysis was restricted to April, while the justification for choosing this month is well communicated, expanding the analysis to the preceding and subsequent months would have added greater depth to the analysis and possibly identified lags and different regional phases of impacts.

Response: *We have followed the reviewer advice and, as mentioned above, we have acquired additional data and extended our study timeframe (i.e. January - June for both 2019 and 2020). This is a major update to the work, and while it has necessitated a great deal of extra work, it has greatly improved the utility of the study.*

My more specific concerns are addressed by line below:

-L19: 'which sectors are most affected, in which regions, and for how long' I am concerned by the for how long part of this statement. Has the study been run sufficiently post covid to capture the 'how long' part of this statement?

Response: *We acknowledge this comment from the reviewer. In our first version, we showed the recovery of some sectors using time series data from the Western Mediterranean only. The extension*

of our dataset from January until June now offers new insights at global scale and even longer in the Mediterranean. The time frame is adequate to capture short-term responses after the first wave of COVID-19. Similarly to the previous economic crisis, however, there will be long-term effects that will need to be assessed in future studies.

-L56: should use caps on Automatic Identification System?

Response: *Changed to caps (L58)*

-L69: Western Mediterranean Sea. The location seems to be driven by access to high level AIS data, if possible expansion to the whole Mediterranean would be appreciated to capture movement of traffic into and out as well as within.

Response: *Access to high resolution AIS data from the Western Mediterranean has been granted by SOCIB. This unique dataset has proven useful to: (1) capture short-term responses due to lockdown effects, (2) provide more details about sectors not considered in commercial products (e.g. recreational vessels). While we acknowledge that expanding the analysis to the whole Mediterranean would be an important contribution, such a dataset is not currently available. Conversely, we have updated the timeframe of the Western Mediterranean dataset until 30th November to provide new insights after the national lockdown periods in the study area (see Fig 8).*

-L71: 'most impacted', I suggest use term 'heavily impacted'. The outbreak is still underway and is increasing in some European countries. How do you quantify and qualify 'impact' in this case? Number of cases, deaths, GDP, etc. all of these are sensitive to country size as well as a host of other factors?

Response: *Changed to "heavily impacted" (L82)*

-L84 (also L98, L107 etc): "Global marine traffic in April 2020 was present in nearly 76.9% of the ocean". Such statistics are picked up and quoted by media without understanding the full meaning and such high percentages are only achieved because of the relatively large grid area used in analysis. This leads to a fallacy when comparing the footprint of a vessel track (the width of a vessel or its operations, along the length of its track) to much larger grid cell. See detailed comments of Amoroso et al. in Science 2018 on global estimates of fishing supplied in Kroodsma et al., 2018. Use of 0.25 deg grid data in the present study fits within the range of areas estimated. Lower end of scale used in fig 2a, 0.001 vessels per km² implies most of cell is empty of traffic, and 80% cells have a density of < 0.0042. The above statement is therefore misleading, especially when moved out of context. <https://science.sciencemag.org/content/361/6404/eaat6713/tab-pdf>. "We accessed the 0.01° grid fishing data made available by Global Fishing Watch (2) and reanalyzed these data at resolutions of ~3100, ~123, and ~1.23 km² (corresponding to 0.5°, 0.1°, and 0.01° at the equator), giving footprint estimates of 49%, 27%, and 4% of ocean area, respectively. Thus, higher-resolution analyses reduced their global fishing footprint estimates by a factor of >10." Use of absolute numbers is more accurate.

Response: *We acknowledge the suggestions pointed out by the reviewer, and we have removed our relative estimates of shipping occupancy in relation to total ocean area. We have limited the use of occupancy as a relative metric for monthly comparisons from the baseline year for each ship category. Such approach is similar to previous works using a similar dataset to describe the seasonal variability of shipping in the Arctic (Eguiluz et al. 2016, DOI: <https://doi.org/10.1038/srep30682>).*

-L99 / L103 Global average decrease of 3.3%, 9.4% relative decrease. Is this L%. If not, why use L% elsewhere and not here. If so, make clear it is L%.

Response: *We have replaced log change (%L) by the conventional relative change in order to avoid confusion in the reporting of results.*

-L174 “Electronic Vessel Monitoring Systems” Up to now we have focussed on AIS, a vessel monitoring system (VMS) has a specific meaning in a fisheries context, suggest keep as AIS
Response: Replaced “electronic vessel monitoring systems” by “AIS data”

-L175 China’s EEZ. Happy to be corrected, but my understanding is China never instigated a national lockdown. The majority of china which represents ~19% global GDP remained open.

Response: We acknowledge the reviewer’s point here. China’s restrictions were not nation-wide and its implementation was uneven across provinces and cities (Ren et al. 2020, DOI: <https://doi.org/10.1080/15387216.2020.1762103>). In order to provide a wider concept of lockdown, not only restricted to the country level, we have removed “national” from the sentence. We maintain our analysis at the EEZ level. In the particular case of China, we believe our approach is adequate given that: (1) China’s lockdown effects were particularly intense in eastern China (Huang, DOI: [10.1093/nsr/nwaa137](https://doi.org/10.1093/nsr/nwaa137)), and (2) there were national-level disease prevention measures (e.g., all cities extended the Spring Festival holiday, required social distancing and urged people to stay at home; He et al. 2020 DOI: <https://doi.org/10.1038/s41893-020-0581-y>).

-L197: ‘240-1,209% by 2050’ unclear what is meant by this. Typo?

Response: This is the range of increase of marine traffic by 2050 estimated by Sardain et al. 2019 across different scenarios (number of vessel movements in comparison with 2014). We have rephrased it as “forecast increases of the global shipping network up to 240-1,209% by 2050 under different economic scenarios”. (L244)

-L201: regional shutdown And Europe. There appears to be an increase in traffic out of Gulf of Aden. Is this coming through from the med via Suez? Some focus on impacts through other trade “pinch points” such as Suez and the Panama canal, and well as Strait of Malacca would also be appreciated.

Response: Great point and maps of absolute differences and contribution per ship category indicates that the increase of marine traffic density within the Gulf of Aden was attributed to tanker vessels (Fig 3). Following reviewer suggestions, we have conducted a new analysis and summarized differences in 10 selected “pinch points” (we use the widely used term “maritime chokepoints”).

-L214: ‘is less likely to be affected than the more vulnerable small-scale fisheries sector that dominates fisheries in many lower-income countries’ The authors should note at this stage that they are unable to monitor these particular fisheries through AIS (it is not referenced until L240). It is mentioned later but relevant here... e.g. this study was not able to assess impacts to small scale fisheries due to limitations of AIS, but studies elsewhere suggest these fisheries are likely to be particularly vulnerable.

Response: We have rephrased and incorporated an statement about the limitation of monitoring small scale fisheries with AIS data. (L231-235).

-L229 ‘a recent regulation....’ Can a reference be provided for this, or more detail on what the regulation mandates in terms of AIS?

Response: Reference to the regulation is now provided, and more detail about the type of measure (i.e. enforcement of the use of AIS for all vessels) is provided (L253-256).

-L240 limitations of AIS data – it would be helpful if authors provided this information at the outset of results and discussion.

Response: We have provided this information earlier in the manuscript to acknowledge the limitation of the AIS. We have mentioned it even earlier, in the introduction when we provide the first background about AIS data (L68).

-L242: 'In this study, we used gridded density maps at the finest data resolution (0.25 degrees) available at global scale and provided on an operational basis'. See previous points on how % and vessel density is presented. Again it would be helpful to present this information earlier.

Response: *We have presented this information at the outset of the results (L98).*

-L244: 'There are additional characteristics that could be derived from raw AIS data (e.g. port calls, individual vessel trajectories) that warrant further attention' You could reference this statement from references already in your list to show that this has can already be done. What you are specifically suggesting is that this is done with respect to COvid 19 e.g. 22, 23 etc.

Response: *We have rephrased this statement and referenced citations presented earlier (L264-266).*

-L253: 'there is, as yet, no international body providing open access to shipping tracking data at a global level'. While we would all like such data to be free, for me this comment misses the advantages that commercial providers have taken the risk of putting constellations of receivers into space at considerable expense and risk and must therefore operate a viable commercial offering to maintain the data service. For the data to be provided for free would therefore require an international body to adequately recompense the providers. Can the authors make a more compelling commercial argument on the cost/benefit of providing such data free as a service to the international community?

Response: *We acknowledge this thoughtful comment from the reviewer and have rephrased this sentence. We agree that commercial interests have facilitated the satellite network deployed. In addition to the previous example from EMODnet, we have included the new UN Global Platform AIS Task team, and focused on the potential use of these emerging datasets (L272-276).*

-L262 'Previous economic crises....' Sentence reads very long. Possibly a missing semi colon, or split into two.

Response: *We have rephrased the sentence (L293-295).*

-L268 'congruent with' I suggest something more specific, such as 'aligned in time with two focal studies'.

Response: *We have rephrased the sentence accordingly (L295-298).*

-L269 'resulting in' suggest reword to something such as that document

Response: *Replaced "resulting in" by "documenting" (L298)*

-L270 'respectively' not needed as reference goes with example.

Response: *Removed "respectively".*

-L272 'our results...could benefit from reduction in marine traffic.' I fail to see how this linkage to MPAs has been shown in the results?

Response: *Decreases in marine traffic found around Galapagos Islands or other coastal areas, like the Mediterranean, where most marine protected areas are located, suggest that human pressures derived from ship-based activities would be lower. We have rephrased the sentence. (L300-303)*

-L273: unclear where this comment on reduction of surveillance effort comes from. What types of surveillance.

Response: *We have rephrase as follows "Conversely, if an associated reduction of surveillance effort by maritime authorities occurs, this could present a higher risk of illicit activities (e.g. illegal fishing, trafficking of drugs), especially in lower-income countries^{48,60}". (L300-305)*

-L289: A previous comment suggested the need for more regional and sectoral focused studies, while this one calls for a more global approach. For me this paper does a good broad brush job of capturing the global but will have inevitably missed more local or focussed impacts. I suggest the focus should be more on the former, or you should be more specific about the types of global study and integrated thinking that are needed.

Response: *We have rephrased the sentence to highlight the need for regional and longer-term studies. (L322)*

-L363. Greater background should be provided on the logarithmic percentage change metric beyond citation. As discussed above I think this is perhaps needed more for the results and interpretation than here in the methods.

Response: *We have replaced logarithmic percentage change metric by percentage change.*

-L368 ABNJ. Occasionally in tables/figures use the term High Seas. Correct, but perhaps confusing and unnecessary. I suggest you use a single term ABNJ.

Response: *Note that summary of the results by ABNJ have been removed from the current version in order to make room for new temporal analyses.*

-L380: 2019 reference. Is there any effort to assess general variability around the 2019 reference point? E.g. How variable was 2019 compared to 2018, 2017 etc. How does the 2020 period compare to background variance? Just looking at the pre-covid 2020 data there remain some periods for some vessel types that were quite different (granted not as large).

Response: *The selection of 2019 is the result of several limitations in AIS. The first point is that global coverage using the same network of satellites is only available from 2019. For example, see previous works (Kroodsmas et al. 2018, DOI: 10.1126/science.aao5646) that found uneven coverage. They used a dataset with 11 satellites, while in our case we use 65. The second point is related with the utilisation of AIS. Adoptions of AIS in recent years is similar for merchant vessels across the world, but different for other categories (fishing, recreational) where it is generally increasing. Fig S6 illustrates this well for the Western Mediterranean, except for large fishing vessels which have required these in EU waters since 2014. Moreover, longer-term datasets used in this study (Fig S6, Fig S9) does not suggest the presence of anomalies in 2019. Overall, the selection of 2019 as reference year is common in the recent literature (e.g. Zheng et al. 2020, DOI: 10.1126/sciadv.abd4998; Liu et al. 2020, DOI: 10.1038/s41467-020-18922-7). We have added all these considerations in the Methods section.*

FIGURES & TABLES:

-Fig.3. ABNJ should read ABNJ sub-regions

Response: *Note that summary of the results by ABNJ have been removed from the current version in order to make room for new temporal analyses.*

-Table S1 – see previous comments on concern calculating % area based on a given grid cell. Occupancy columns therefore concern me.

Response: *As mentioned earlier, we have replaced %L by standard % change. Moreover, given the new temporal dimension of the datasets, we have replaced the table by a figure (Fig 4).*

-Table S2: Standard date format of dd-mm-yyyy preferred.

Response: *We have removed previous Table S2 in the new version of the manuscript, and incorporated maximal and median drop values into Figure 8.*

-Fig. S2 Here and several similar figures. No units on color bar or in legend. Is colour scale linear-presumably logarithmic as in Fig 2a? It would be helpful to have some additional ticks, and scale values.

Response: *In order to clarify and harmonize across similar figures we have followed the same approach: replaced natural logarithm transformation by log base 10 transformation and added a tick at every power of 10 (e.g. Eguiluz et al. 2016).*

-Fig. S3 It would be helpful to have some additional ticks, and scale values.

Response: *We have removed this figure in the current version of the manuscript.*

-Fig S4: y axis should show L% not %

Response: *We have removed this figure in the current version of the manuscript.*

-Fig. S5 x axis should show L% not %

Response: *We have removed this figure in the current version of the manuscript.*

-Fig S7 should read ABNJ Sub regions

Response: *We have removed this figure in the current version of the manuscript.*

-Fig. S11 L% on y axis, ROI on inset globe difficult to read

Response: *We have replaced a globe inset with a new one that makes a zoom into the Western Mediterranean (now Fig8).*

-Supplementary Data tables 1-4 replace column header with L% rather than %.

Response: *We have removed Supplementary Data tables in the current version of the manuscript.*

Reviewer #2:

This paper does an excellent job quantifying changes in vessel activity related to COVID-19 lockdown measures. I appreciated seeing these values displayed in space, time and by ship type, as well as zoomed into the Mediterranean for more detailed analyses. The Stringency index is quite useful as a metric for contrasting ship movements to the intensity of governmental mitigation measures to COVID-19, such as lockdowns. This paper is particularly relevant to the marine ecological and blue economy fields as a baseline for subsequent study on how shipping may impact ecological status and other economic activities in the context of assessment before, during and after COVID-19 as a natural experiment. Towards this aim, I am especially pleased to see the free availability of the spatial and tabular datasets along with the R code for conducting the analysis and visualization.

***Response:** Many thanks for such a positive and encouraging review.*

I recommend this manuscript for publication as it lucidly presents its novel findings unique to our COVID times as well as relevance for further study across the marine scientific community. Nonetheless I would like to share some additional minor suggestions for improvement.

Wording:

Replace “ABJN” (lines 368, 588) with “ABNJ” (Areas Beyond National Jurisdiction)

***Response:** Note that summary of the results by ABNJ have been removed from the current version in order to make room for new temporal analyses.*

Replace “vessel monitoring systems” (line 175) intended generically here with another term so as to not confuse the AIS data used in this study with that from a separate and different ship data type called “vessel monitoring system” (VMS) data (e.g. see <https://globalfishingwatch.org/vessel-tracking-data/>)

***Response:** We have replaced “vessel monitoring systems” by “AIS data” in order to avoid confusion and be more specific to the source of information used for our analysis.*

The phrase “entered into force” (line 229) seems odd, so could be replaced with “became enforced”.

***Response:** In combination with Reviewer#1 suggestions, we have rephrased this sentence as “national regulation that enforced the use of AIS for all vessels by the 20th August 2019 (Ministerial Regulation PM 7/2019)” (L253-256).*

Although described in the Methods, some mention in the Introduction of the shift from the AIS receivers being on land to satellite would help clarify how global monitoring is technically feasible in the sentence at lines 56-7.

***Response:** Incorporated a comment in the introduction to highlight the relevance of S-AIS (L58).*

Similarly in the Introduction, I would like to see explicit mention of how AIS is generally limited to bigger ships (e.g. > 300 tons) that is well described in the Methods.

***Response:** We have moved the main introductory paragraph about AIS from the Methods to the Introduction, thus providing a clearer description of AIS (L62-67).*

Methods:

Can you produce a direct correlation between Stringency Index and shipping activity by country?

This would presumably be done spatially within a country’s surrounding EEZ waters (versus the nationality of ship origin). This could be assessed at various time lags. This would then provide a coefficient (for instance from a linear model) predicting the amount of shipping reduction resulting from the level of Stringency with a measure of uncertainty.

Response: We really appreciate this suggestion. While the available AIS datasets do not provide information about the nationality of the ship origin, we have analysed the changes within EEZs in relation to the confinement measures using linear mixed models (LMMs). We included the relative change as the dependent variable and added fixed effects of Stringency index and economic classification from the World Bank. We included Country as a random effect, which helps to account for spatial autocorrelation and internal factors not incorporated in the model. We found a significant effect of the confinement measures on the change of marine traffic for all vessel categories, except for fishing vessels (Table S1, Fig S5). In addition, we found that the effect of confinement measures was uneven across economies, with lower-income countries being less affected by confinement measures.

The comparison of global marine traffic between April of 2019 and 2020 is quantitatively illustrative of the reduction (1.4% global occupancy), but I am also curious to know how April 2020 compares with the expectation of April 2020 marine traffic without COVID if it continued with its expected growing trend -- a presumably bigger difference.

Response: We agree with the comment. Acknowledging the lack of long-term AIS dataset available to calculate a forecast for 2020, we have used a long-term dataset of ship density derived from altimetry (Tournadre 2014, DOI: <https://doi.org/10.1002/2014GL061786>). We provide details on the procedure on Supplementary methods and included one comment in the discussion about the underestimation of changes (L239-242).

I appreciate the finer scale Mediterranean analysis beyond the global. This also begs the question of how much more information are you getting from using the terrestrial versus satellite data? Is there a simple comparison between the two over the same area that you could make? For instance, the denser the number of AIS signals in busy shipping areas, the higher the dropout rate for the satellite receiver. Mention of these shortcomings in the global dataset justifies this more detailed view.

Response: T-AIS from the Mediterranean has proven useful to: (1) capture short-term responses due to lockdown effects on a daily basis, and (2) provide more details about sectors not considered in the global dataset (e.g. recreational vessels). We have conducted a new comparison between T-AIS and S-AIS (see Fig S8). We have reprocessed the T-AIS data to derive a gridded product at the same resolution of S-AIS (0.25 degrees). Then, we have compared them by calculating the absolute difference between equivalent months. Such analysis supports our initial hypothesis: there is an underestimation of traffic density by T-AIS in (1) areas with a lower density of antennas (i.e. north Africa), and (2) in areas furthest from the coast (e.g. ocean area between Balearic Islands and Sardinia). Moreover, such underestimation is larger during winter months, when AIS detection range could be reduced by adverse metocean conditions. These results are in agreement with a recent study that used T-AIS (Holmes et al. 2020, DOI: <https://doi.org/10.1371/journal.pone.0230494>), and support our decision to restrict our regional analyses to coastal areas (<24 nm) from EU countries. Overall, results suggest that S-AIS is most suited to capture the spatial variability across large regions. Furthermore, note that the new constellations of satellites have been expanded recently (65 nano-satellites, this study; 11 satellites, Kroodsmas et al. 2018, DOI: [10.1126/science.aao5646](https://doi.org/10.1126/science.aao5646)), and now offer a better detection capacity.

Discussion:

Although some relevant examples in marine traffic were mentioned, it might be interesting to note how other 'natural experiments' have influenced our understanding of ecological impacts and societal decision-making. For instance, how did 9/11 and the momentary cessation of plane traffic influence our ecological understanding of planes on birds?

Response: We have incorporated further examples from previous disruptions (e.g. aircraft effects on temperature daily range uncovered after 9/11 disruption, Travis et al. 2020, DOI: [10.1038/418601a](https://doi.org/10.1038/418601a))

and added new COVID-19 examples related with reduction of noise (Lecocq et al 2020, DOI: 10.1126/science.abd2438) and its effects on birds (Derryberry et al. 2020, DOI: 10.1126/science.abd5777). (L278-287)

Reviewer #3:

Summary

The manuscript uses AIS tracking data from ships to characterise changes in the density of global shipping at varying spatial scales. The Stringency Index was used to indicate when lockdown measures were at their highest, in April 2020. AIS data were sourced from satellites (S-AIS) and terrestrial receivers (T-AIS). S-AIS data were analysed in April 2019 (as a comparator benchmark) and April 2020, and the results were compared through cell-by-cell subtraction at different spatial scales of grid cell, EEZ, ABJN, marine ecoregions, and FAO major fishing areas. T-AIS data from the Mediterranean Sea were analysed from 2016-2020 to show how the number of vessels changed over time. The global results give a snapshot in time of how vessel density changed at different spatial scales.

General

The writing and figures in the manuscript are of high quality, and clearly a lot of time and thought has been put into producing the set of images, which are easily interpretable and well presented.

Response: Many thanks for such a positive and encouraging review.

However, I do not think the manuscript, as it is now, contributes to the scientific knowledge base sufficiently enough to warrant publication. Set out below are the key areas that the manuscript needs to address to become publishable by Nature Communications.

1. Stringency Index – this was in a Europe-centric context. The authors use the rationale that the majority of Europe had the most stringent lockdown measures in place in April 2020, which seems plausible given Fig. 1. However, global maps of vessel density are then presented at this time point, at a time when many countries that have a high number of vessels associated with them (e.g. China, Norway, Singapore, Norway, USA, S. Korea, Denmark), and/or have significant fishing industries (e.g. China, Indonesia, USA, Philippines, Norway, S. Korea), were not in the most stringent lockdown. The Europe-centric view not only limits the interpretation of the global maps that have been produced, but undermines their validity because some of the interpretation is discussed in relation to areas that were not in full lockdown at the time (Fig 2d, e, f, h). This must be accounted for in the analysis. It is reasonable to assume that countries that were in lockdown would have reduced vessel traffic and countries that were not in lockdown or in a less severe lockdown would have increased or similar traffic. What this analysis does not do is show how this trend plays out when taking into account the underlying restrictions over time.

Response: This was a good point, well made and we have now expanded the S-AIS dataset from the month of April. Now our study spans 6 complete months from January until June. Such an extension allows us to consider the variability in restrictions over time. In particular, we have conducted a new analysis to assess the effect of confinement measures per country accounting for the temporal variability in both traffic and stringency index. We thank the reviewer for this, making these changes, has taken a lot more work but has transformed the utility of the manuscript.

2. The analysis of cell-by-cell subtraction does not take account of dependency between the grid cells. Vessel densities are bound to be spatially correlated in that grid cells near each other are more similar. Without accounting for this, a cell-by-cell subtraction is likely to be biased. How do you know whether a change is significant? There is literature on comparing spatial differences such as Hagen-Zanker (2006), Moran's I or Geary's C tests (Cliff and Ord (1970), or Environmental Niche Models (Warren et al, 2008).

Response: We acknowledge the potential effect of spatial autocorrelation in statistical analysis derived from the gridded data. The spatial distribution of marine traffic along shipping lanes may lead to violation of the basic assumption of independent errors in standard statistical models. We

have followed these considerations when analysing the data. (1) First, for cell-by-cell subtraction, we do not conduct a statistical test of significance. Similarly to previous works (e.g. Kroodsma et al. 2018, DOI: 10.1126/science.aao5646), we have addressed a descriptive approach by quantifying the magnitude of change. Our aim is to provide the baseline data required for conducting further studies related to the change of traffic density. For example, other groups who would like to conduct a regression with changes in underwater noise. In such models, care should need to be taken if there is presence of spatial autocorrelation in the residuals. (2) Second, we have considered the potential issue of spatial autocorrelation when designing the new statistical analysis on the effect of the Stringency Index. In this case, we have aggregated grid cells by EEZ and defined the country random effect to control for the spatial structure in our sampling units. (3) Finally, we also considered smoothing the densities maps using a focal window. However, we rejected that approach as the location error of the GPS is much lower than the grid size, thus reducing the potential noise that we could find at finer resolutions.

3. Limitations – there are caveats and limitations around using AIS data, and some of these are mentioned (Line 240). However, there are a number of other caveats, for example, locations from fishing vessels may not be as accurate as other vessel types because AIS on these vessels can be activated manually.

Response: We have incorporated the issue of transmission gaps in fishing vessels in the AIS limitations section (Ferrà et al. 2020, DOI: 10.3389/fmars.2020.580612). Note that we have moved our comment about limitations in the introduction (L68-69).

The manuscript mentions few antennas for T-AIS in north Africa (Line 355) and temporal bias was reduced but was there also spatial bias due to non-homogeneity of antenna, and if so, how can this be accounted for?

Response: We have conducted a new comparison between T-AIS and S-AIS in order to assess potential spatial and temporal biases in the T-AIS dataset (see Supplementary Methods and Fig S8). This analysis supports our initial hypothesis: there is an underestimation of traffic density by T-AIS in (1) areas with a low density of antennas (i.e. north Africa), and (2) in areas farthest from the coast (e.g. ocean area between Balearic Islands and Sardinia). Moreover, such underestimation is greater during winter months, when AIS detection range could be reduced by adverse metocean conditions. Overall, these results support our initial approach used to consider spatial bias in T-AIS: restricting our analysis to coastal areas (<24 nm) of EU countries, with larger density of land-based AIS stations.

Specific points

Line 26 – Guidance of ‘potential effects of COVID-19’ have not been demonstrated in this manuscript.

Response: Thanks for pointing out this lack of clarity. We have modified the text to read as follows, “Our approach provides guidance for large-scale monitoring of the progress and potential effects of COVID-19, or other global shocks, on vessel traffic that may subsequently influence the blue economy and ocean health.” - as noted by Reviewer 2. (L26-28)

Line 73 - Although socio-economic and environmental effects of COVID-19 are mentioned, it is not clear how measuring changes in vessel density can provide guidance on these effects? How can this information be linked explicitly?

Response: Thanks for pointing out this lack of clarity. We have tried to make these linkages clearer. **Socio-economic effects:** We highlight that the effects have been highly variable in space, time and among sectors. We show that one of the ship types with a major decrease are passenger vessels, particularly in the Mediterranean. **Environmental effects:** These are yet to be fully elaborated, our maps show a good correspondence with previously described environmental effects due to COVID-19.

They include decreases in underwater noise in Port of Vancouver and improvements in water transparency in Venice. Unlike other port-based indicators (e.g. port calls) our approach provides a spatially-explicit assessment of changes which has the potential to be linked to environmental pressures.

Line 150 – This is only the case because of the time point (April 2020) chosen for the analysis. If the time point was earlier in the year when China was locked down or later when the US / South America, and Africa had more severe restrictions, shipping activities, and therefore change in density, would look very different.

Response: *We have now extended both the global and regional datasets, so now provide a radically better representation of the temporal changes.*

Line 184 – This is solely because shipping traffic aggregates close to the coast.

Response: *Agreed. We have removed this sentence. Also, note that in the current version we have decided to remove the summaries by marine ecoregions to allow us to focus on the temporal extension of global and regional coverage.*

Line 186 – You must caveat this because this is partly due to the choice of time point (April 2020) and you found the Mediterranean Sea is particularly affected because of the fine-scale analysis you undertook.

Response: *After the extension of the global dataset, we can see that European Seas were heavily impacted during the first half of 2020 (Fig 3, Fig 5a).*

Lines 202-203 – This is the first time the Silk Road and Strait of Malacca has been mentioned – where is this analysis?

Response: *For better clarity, we have removed the reference to the Silk Road. But, as also suggested by Reviewer#1, we have conducted a new analysis and summarized differences in 10 selected “maritime chokepoints”, which includes the Strait of Malacca (Fig 6).*

Line 205 – Where is the analysis that shows an increase in demand for oil tankers? Fig. S11 shows smaller increases and larger decreases.

Response: *We have incorporated new references to support this statement. COVID-19 pandemic has cut oil demand (Jefferson et al. 2020, DOI: <https://doi.org/10.1016/j.erss.2020.101669>). However, tankers that transport crude oil were found not to be affected in terms of port calls, supporting the importance of using tankers as storage capacity during periods of oil market distress (Michali et al. 2020, DOI: <https://doi.org/10.1016/j.trip.2020.100178>).*

Line 289 – Rerunning the analysis from different time points would provide different snapshots but how could that then be used to give facilitate insight into long-term impacts, and how does demonstrating changes in vessel density relate to anthropogenic impacts on the marine environment?

Response: *Changes in vessel density relate with changes in human pressures (e.g. fishing effort, underwater noise), which in turn result in anthropogenic impacts on marine ecosystems (e.g. Halpern et al. 2019, DOI: <https://doi.org/10.1038/s41598-019-47201-9>). For example, previous studies that align with our map of changes in vessel density have reported changes in underwater noise (Thomson & Barclay et al. 2020, DOI: <https://doi.org/10.1121/10.0001271>) or water turbidity (Braga et al. 2020, DOI: <https://doi.org/10.1016/j.scitotenv.2020.139612>). Because such changes varies with time, adding a temporal dimension to our data can further improve the assessment of ship-derived impacts on the marine environment.*

Line 329 – What is the rationale for choosing <100 cells? Is this an arbitrary threshold?

Response: Our initial rationale was to filter out small patches of vessel counts that appeared disconnected and presented anomalies in the data (e.g. exact same speed or vessel counts). We initially used an arbitrary threshold based on the number of cells. After a more in-depth analysis, we have replaced such criteria by an area-based criteria, which accounts for the variability in grid cell size across the latitudinal gradient (see Supplementary Methods).

Line 330 – Remove the word ‘accurate’ or rephrase the sentence. Any projection is accurate for some property, in this case it’s area.

Response: Removed “accurate” from the sentence.

Line 345 – How did you define the additional category of ‘recreational’?

Response: Recreational category was defined as the combination of two categories used by Marinetraffic.com, "Sailing Vessel" and "Pleasure Craft", which correspond to ship types codes 36 and 37, respectively. We have included the type codes in the methods.

Line 369 – ‘We averaged per-pixel values, allowing direct comparison among regions despite large differences in size’ – it would have been more informative to use the standard deviation from those mean values in the comparison

Response: We acknowledge the suggestion from the reviewer. We have estimated both average and standard deviation to summarize monthly changes in 2020 from 2019. Note that previous aggregation using other regions (e.g. ABNJ) have been removed, and we now report both metrics for maritime chokepoints (Fig 6) and EEZs (Fig 7).

Line 371 – This could bias your results. You need to explain how you included them in the analysis if the spatial scales were not the same. The rationale for including the EEZs is questionable – firstly, this analysis is not about air pollution or underwater noise and it doesn’t link to it; secondly stating that environmental pressures are ‘diffuse’ isn’t a rationale.

Response: We acknowledge the reviewer comments on this point. While we are not linking our data with environmental pressures (e.g. underwater noise), we consider that our maps have an enormous potential to be linked to such kinds of environmental assessments in the future. We therefore aimed at summarizing our findings for different regions such as EEZ. We have now conducted a new analysis aimed to assess the effect of confinement measures. We have taken into consideration the reviewer suggestion, and have selected regions larger than 2,307 km² (i.e. equivalent to the size of three grid cells in the Equator) to ensure we had at least three observations per EEZ.

REVIEWERS' COMMENTS

Reviewer #1 (Remarks to the Author):

Having read the response to both my comments and those of my fellow reviewers I am satisfied that these have been addressed in the revised manuscript, or where this is not the case, that sufficient explanation has been offered of the course taken. The authors have gone to great lengths to improve the readability of their article, and I appreciate the improvements and updates to the figures in particular.

I am happy to recommend that this article proceeds to publication and I look forward to seeing the article in print.

Gwilym Rowlands

Reviewer #2 (Remarks to the Author):

Well done with the revision expanding overall analyses in time and chokepoints in space! The authors put in a great deal of additional work responding in full to all the reviewers' comments.

I only have the most modest follow-up suggestion:

L244 "and forecast increases of the global shipping network up to 240-1,209% by 2050 under different economic scenarios." This range of values is still odd to read in the sentence after revising based on suggestion from reviewer #1, even if modifying with proper en dash (-) between numbers. Suggest: "and forecast increases of the global shipping network ranging from 240% to 1,209% by 2050 under different economic scenarios."

Authors' response to Final revisions on NCOMMS-20-28656
"Tracking the global reduction of marine traffic during the COVID-19 pandemic "

RESPONSE TO REVIEWER COMMENTS

Reviewer #1:

Having read the response to both my comments and those of my fellow reviewers I am satisfied that these have been addressed in the revised manuscript, or where this is not the case, that sufficient explanation has been offered of the course taken. The authors have gone to great lengths to improve the readability of their article, and I appreciate the improvements and updates to the figures in particular.

I am happy to recommend that this article proceeds to publication and I look forward to seeing the article in print.

Gwilym Rowlands

***Response:** Many thanks for such a positive recommendation.*

Reviewer #2:

Well done with the revision expanding overall analyses in time and chokepoints in space! The authors put in a great deal of additional work responding in full to all the reviewers' comments.

I only have the most modest follow-up suggestion:

L244 "and forecast increases of the global shipping network up to 240-1,209% by 2050 under different economic scenarios." This range of values is still odd to read in the sentence after revising based on suggestion from reviewer #1, even if modifying with proper en dash (–) between numbers. Suggest: "and forecast increases of the global shipping network ranging from 240% to 1,209% by 2050 under different economic scenarios."

***Response:** Many thanks for such a positive and encouraging feedback. We acknowledge the suggestion from the reviewer. We have replaced "and forecast increases of the global shipping network up to 240-1,209% by 2050 under different economic scenarios." by "and forecast increases of the global shipping network ranging from 240% to 1,209% by 2050 under different economic scenarios." (L245).*